# HUMAN-OBJECT INTERACTION VIA AUTOMATICALLY DESIGNED VLM-GUIDED MOTION POLICY

**Zekai Deng**[1]   **Ye Shi**[1,2]   **Kaiyang Ji**[1]   **Lan Xu**[1]   **Shaoli Huang**[3]   **Jingya Wang**[1*]
[1]ShanghaiTech University    [2]InstAdapt    [3]AgiBot
{dengzk2023,shiye,jiky2024,xulan1,wangjingya}@shanghaitech.edu.cn,
shaolihuang@agibot.com

## ABSTRACT

Human-object interaction (HOI) synthesis is crucial for applications in animation, simulation, and robotics. However, existing approaches either rely on expensive motion capture data or require manual reward engineering, limiting their scalability and generalizability. In this work, we introduce the first unified physics-based HOI framework that leverages Vision-Language Models (VLMs) to enable long-horizon interactions with diverse object types — including static, dynamic, and articulated objects. We introduce VLM-Guided Relative Movement Dynamics (RMD), a fine-grained spatio-temporal bipartite representation that automatically constructs goal states and reward functions for reinforcement learning. By encoding structured relationships between human and object parts, RMD enables VLMs to generate semantically grounded, interaction-aware motion guidance without manual reward tuning. To support our methodology, we present Interplay, a novel dataset with thousands of long-horizon static and dynamic interaction plans. Extensive experiments demonstrate that our framework outperforms existing methods in synthesizing natural, human-like motions across both simple single-task and complex multi-task scenarios. For more details, please refer to our project webpage: https://vlm-rmd.github.io/.

## 1 INTRODUCTION

Human-Object Interaction (HOI) understanding is fundamental to advancing embodied AI systems in simulation, animation, and robotics. Existing approaches to HOI learning generally fall into two categories. The first paradigm employs motion tracking policies to imitate reference demonstrations (Wang et al., 2023; 2024a; Xu et al., 2025). While these methods can reproduce specific interaction patterns, they face critical limitations: Their heavy reliance on high-quality motion capture data creates scalability bottlenecks, and their strict adherence to reference trajectories inherently restricts their ability to generate novel interactions beyond the training distribution.

The second paradigm adopts a task-centric perspective, developing specialized policies for individual interactions like sitting (Chao et al., 2021; Zhang et al., 2022) or carrying (Xu et al., 2024b; Szot et al., 2021; Deitke et al., 2022; Gao et al., 2024). However, these methods encounter two fundamental challenges. First, they require labor-intensive reward engineering by domain experts — a particularly demanding requirement given the complex dynamics and contact-rich nature of realistic HOI scenarios (Sutton & Barto, 2018). This manual reward design process inherently limits generalizability across interaction types, creating a need for automated objective formulation. Second, constrained by artificially designed, single-objective reward mechanisms, trained policies often overfit to specific behavioral patterns. Producing task-compliant but biomechanically unrealistic motions that violate natural human kinematics.

Recent efforts such as Eureka (Ma et al., 2023) and Grove (Cui et al., 2025) leverage large language models (LLMs) to automatically generate reward function code. However, they rely on iterative search paradigms that are often sample-inefficient and computationally costly. Xiao et al. (2024) advances this line of work by unifying reward design through a "chain-of-contacts" abstraction, which

---

*Corresponding author.

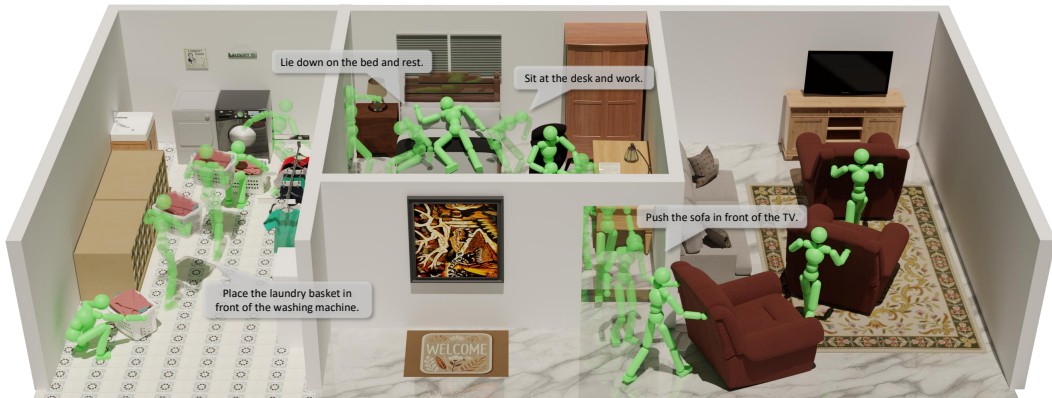

Figure 1: Our framework automatically constructs both goal states and reward functions for diverse interaction tasks in reinforcement learning. By leveraging VLM guidance, the learned motion policy drives physics-based characters to perform coherent, long-horizon interactions with static and dynamic objects, producing natural and task-consistent behaviors.

models an interaction as a discrete sequence of contact events. While conceptually elegant, this formulation overlooks the motion dynamics of interactions and lacks the capacity to model whole-body coordination or accommodate dynamic objects. Consequently, it struggles to handle dynamic interactions and often produces jittery, suboptimal behaviors even in static interaction scenarios.

To address these limitations, we propose a physics-based HOI framework that leverages VLMs to automatically construct goal states and reward functions, guiding the motion policy to perform long-term interactions with diverse objects. Specifically, we aim to empower VLM to provide motion-level, interaction-aware guidance by leveraging its abilities in motion imagination and semantic reasoning. Drawing inspiration from the classical notion of relative motion (Zanstra, 1924), we introduce *Relative Movement Dynamics* (RMD), a structured representation that encodes the fine-grained spatio-temporal relationships between human parts and object parts across an interaction sub-sequence. For example, in the task of lifting a box, the relative spatial configuration between the hands and the box remains stable, providing a natural constraint that captures both discrete interaction goals (e.g., contact) and continuous dynamics (e.g., coordinated movement). RMD is designed to prompt VLMs to imagine interaction dynamics while effectively grounding their high-level reasoning in motion-level patterns. This formulation enables VLMs to move beyond symbolic planning and participate in dynamic skill composition, thereby supporting more expressive and generalizable HOI motion policy learning. Building on RMD, our framework automatically constructs unified goal states that support both static and dynamic interactions, along with automatically designed reward functions that guide motion patterns consistent with the plan.

Our framework supports long-horizon interactions with diverse object types. However, existing datasets typically focus either on static interactions (Xiao et al., 2024) or on object rearrangement (Xu et al., 2024b), and thus fail to cover this setting. To address this gap, we construct a new dataset, *Interplay*, which includes long-horizon static and dynamic interaction tasks across varied scene contexts. This dataset enables systematic evaluation of our framework in a long-horizon, multi-task scenario. Experimental results demonstrate that our method achieves promising quantitative performance and generates natural, human-like motions across a diverse range of HOI tasks.

In summary, our contributions include:

We propose the first unified physics-based HOI synthesis framework for long-horizon human-object interactions leveraging the powerful world knowledge of VLMs, supporting a diverse range of objects, including static, dynamic, and articulated ones.

We introduce VLM-Guided Relative Movement Dynamics (RMD), a fine-grained spatial-temporal bipartite diagram to automatically construct goal states and reward functions for reinforcement learning. This approach bypasses manual reward engineering while supporting diverse HOI, including both static and dynamic interactions.

Table 1: Comparative analysis of key features between ours and other methods.

| Methods | Articulated Object | Automated Reward Design | Dynamic Object | Long-horizon Transition | HOI Guidance | Multitask | High-level Planning | Unified Policy |
|---|---|---|---|---|---|---|---|---|
| InterPhys Hassan et al. (2023) | | | ✓ | | Coarse | | | |
| InterScene Pan et al. (2024) | | | | Rule-based FSM | Coarse | | | |
| TokenHSI Pan et al. (2025) | | | ✓ | Rule-based FSM | Coarse | ✓ | | ✓ |
| UniHSI Xiao et al. (2024) | | ✓ | | ✓ | Coarse | ✓ | ✓ | ✓ |
| Ours | ✓ | ✓ | ✓ | ✓ | Fine | ✓ | ✓ | ✓ |

We introduce *Interplay*, a novel dataset comprising thousands of interaction plans that cover long-horizon static and dynamic interaction tasks. Extensive simulation results demonstrate the effectiveness of our approach on both single-task and long-horizon multi-task scenarios.

## 2 RELATED WORK

**Human Motion Synthesis with Static Scenes.** In the field of character animation and motion synthesis, most research has focused on human motion interacting with static 3D scenes (Li et al., 2019; Zhang et al., 2020; Xuan et al., 2023; Zhao et al., 2022; Lee & Joo, 2023; Wang et al., 2022b; 2024b; Starke et al., 2019; Zhao et al., 2023a). Typically, researchers break down complex instructions into several key static poses within the scene (Hassan et al., 2021; Wang et al., 2022a), and then use motion inpainting and optimization techniques to generate transitions between these key poses (Wang et al., 2020). However, this approach often leads to mediocre and inconsistent motions due to the limited expressiveness of the intermediate sequences. Recently, diffusion-based methods like (Ho et al., 2020; Huang et al., 2023; Tang et al., 2024; Yi et al., 2025; Zhao et al., 2024; Tevet et al., 2024) have achieved better results in human-scene synthesis. However, these data-driven approaches are limited to generating short-term motions due to constraints in the dataset, and the physical plausibility of the generated motions is not guaranteed. Apart from data-driven kinematic approaches, some studies have explored the problem within a reinforcement learning framework. For instance, (Zhao et al., 2023b; Zhang & Tang, 2022) achieved long-term goal-oriented behaviors by leveraging motion primitives with specific reward designs. Xiao et al. (2024) decomposes instructions into a sequence of point-reaching control tasks under a unified formulation. However, it addresses only the spatial constraints of static scenes while overlooking the temporal dynamics of objects, rendering it inadequate for managing complex dynamic interactions.

**Kinematic-based Human Motion Synthesis with Dynamic Objects.** Research in human motion synthesis has increasingly focused on modeling interactions with dynamic objects (Starke et al., 2020; Jiang et al., 2022; Zhang et al., 2022; 2024). Diffusion-based frameworks (Li et al., 2023b;a; Pi et al., 2023; Peng et al., 2023; Ji et al., 2025; Zhang et al., 2025), guide motion generation with object trajectories but often lack realism due to predefined object paths that fail to account for physical plausibility. In contrast, InterDreamer (Xu et al., 2024a) first generates human motion and subsequently uses a pre-trained world model to produce object trajectories, though this approach is limited by the simplicity of the world model, leading to inaccuracies in trajectory prediction. Some studies attempt to address these limitations by jointly modeling human-object interactions with supplementary guidance techniques, including relation intervention (Wu et al., 2024a), contact prediction (Diller & Dai, 2024), and affordance estimation (Peng et al., 2023). Truman (Jiang et al., 2024b) and Lingo (Jiang et al., 2024a) trained on a high-quality human-scene interaction (HSI) dataset, achieves dynamic stability through an auto-regressive diffusion model guided by action labels or text. Nevertheless, kinematic models continue to face challenges such as penetration, sliding issues, and difficulties in generating long-term motion, requiring extensive annotations.

**Physics-based Human Motion Synthesis with Dynamic Objects.** To generate physically plausible motions, reinforcement learning methods have been shown to effectively train HOI skills using motion capture data (Chao et al., 2021; Merel et al., 2020; Peng et al., 2018; Xie et al., 2023; Xu et al., 2025; Wu et al., 2024b). AMP (Peng et al., 2021) introduced an adversarial motion prior framework for realistic motion synthesis. InterPhys (Hassan et al., 2023) further extended this framework by incorporating an HOI motion prior, achieving success in tasks such as sitting, lying, and carrying. Further advancements have led to successful applications in sports activities, including basketball (Wang et al., 2023; 2024a), tennis (Zhang et al., 2023), skating (Liu & Hodgins, 2017), and soccer (Luo et al., 2024; Liu et al., 2022; Xie et al., 2022). TokenHSI (Pan et al., 2025) integrates

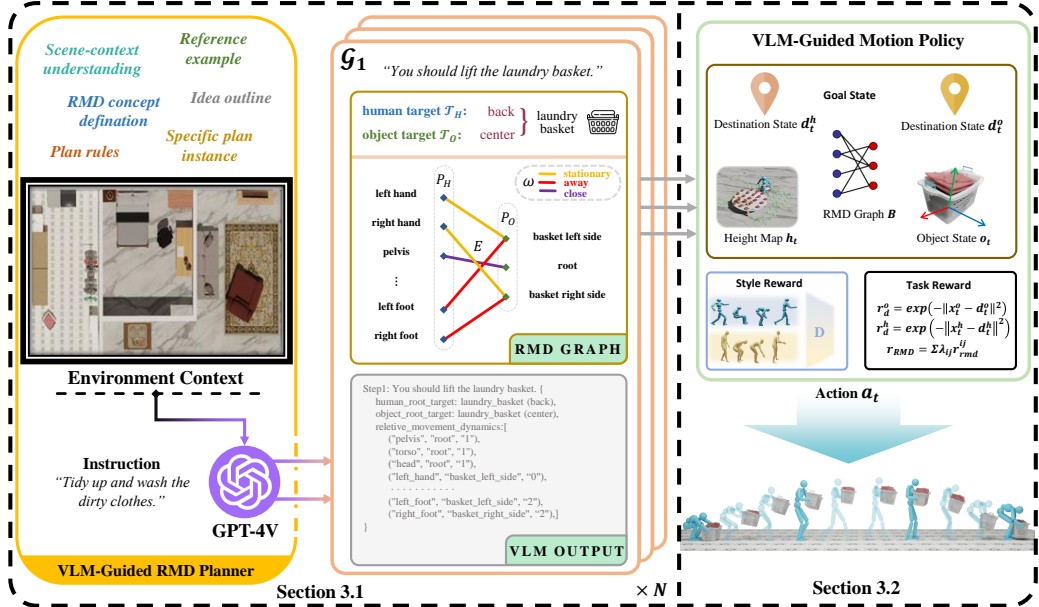

Figure 2: **An overview of our architecture.** Receiving instruction and environment context as input, the VLM-Guided RMD Planner generates a multi-step interaction plan in the form of RMD. Based on this plan, our framework automatically designs both goal states and reward functions, enabling the VLM-Guided Motion Policy to execute the interaction step by step.

multiple skills into a single policy via a task tokenizer. These approaches often depend on heuristic designs and struggle to generalize to longer, multi-round scenarios. This limitation highlights the need for a principled representation that can bridge short-horizon rules and long-horizon interaction dynamics. Our method seeks to address these limitations by introducing a Relative Movement Dynamics representation, enabling the model to capture both spatial and temporal dynamics effectively. This results in physically plausible and temporally coherent interactions, eliminating the need for manual annotations and enhancing realism in dynamic, long-term HOI tasks. The detailed comparisons of key features are listed in Tab. 1.

## 3 METHOD

Our method comprises two tightly coupled components. First, we describe how a VLM is leveraged to translate high-level instruction and environment context into a sequence of structured interaction plans. This process is grounded in our proposed concept of RMD, which models fine-grained spatio-temporal relationships between human and object parts and serves as the foundation for interaction planning (Sec. 3.1). Then, we present a VLM-guided motion policy learning framework, where each RMD plan is automatically transformed into goal state and a corresponding reward function, enabling the agent to learn executable motion policies without manual specification (Sec. 3.2).

### 3.1 VLM-GUIDED RMD PLANNER

We employ GPT-4V (Achiam et al., 2023) as our VLM planner to bridge high-level task instructions and low-level interaction execution. As illustrated in Fig. 2, the planner takes three inputs: a high-level textual instruction $\mathbf{I}$, a top-view contextual image of the environment $\mathbf{C}$, and a set of modular prompts designed to guide the decomposition of the task into detailed interaction plans.

Effectively leveraging the motion imagination capabilities of VLMs for HOI planning necessitates a principled formulation of the interaction process. Simplistic abstractions, such as the chain-of-contacts (CoC) proposed by UniHSI (Xiao et al., 2024), is inadequate: by imposing only transient point-contact constraints that are discarded immediately upon satisfaction, CoC fails to capture the evolving spatio-temporal relationships critical for modeling dynamic and coordinated interactions.

In contrast, we propose a novel concept, *Relative Movement Dynamics* (RMD), which encodes fine-grained spatial and temporal relationships between human body parts and object components throughout the course of an interaction. This structured formulation enables the VLM to reason more effectively about the physical organization of human–object interactions and to produce structured, executable plans aligned with underlying motion dynamics. Our key insight is that human-object interactions can be abstracted as the relative motion between two sets of rigid bodies: human body parts and object parts, evolving over time. We formalize this abstraction as a bipartite graph $\mathcal{B}$ that encodes part-level motion dynamics. Let

$$P_H = \{p_{h_1}, p_{h_2}, \ldots, p_{h_m}\}, \quad P_O = \{p_{o_1}, p_{o_2}, \ldots, p_{o_n}\} \tag{1}$$

denote the sets of human and object parts, respectively. Then the graph $\mathcal{B}$ is defined as,

$$\mathcal{B} = (V, E, w), \qquad V = P_H \cup P_O, \quad E \subseteq P_H \times P_O, \quad w : E \to \{0, 1, 2, 3\}. \tag{2}$$

Each edge $e_{ij} = (p_{h_i}, p_{o_j}) \in E$ connects a human part $p_{h_i} \in P_H$ to an object part $p_{o_j} \in P_O$, with an associated weight $w_{ij} \in \{0, 1, 2, 3\}$ that characterizes their relative-motion pattern: $w_{ij} = 0$ indicates stationary contact, $w_{ij} = 1$ denotes approaching motion, $w_{ij} = 2$ represents separating motion, and $w_{ij} = 3$ implies no consistent relative trend.

To facilitate the planner's ability to infer such dynamics, we adopt a modular prompt design strategy. Each prompt segment is crafted to trigger a specific reasoning ability, such as environmental parsing, part-level object understanding, motion dynamic inference, or symbolic representation generation. Through this mechanism, the VLM is guided to produce both the RMD graph $\mathcal{B}$ and two spatial anchors: the target location for the human root $\mathcal{T}_H$ and the target location for the object root $\mathcal{T}_O$, which together define the global interaction objective.

Finally, the planner outputs an interaction plan $\mathcal{D}$ as a sequence of $N$ structured steps,

$$\mathcal{D} = \{\mathcal{G}_1, \mathcal{G}_2, \ldots, \mathcal{G}_N\}, \tag{3}$$

where each step $\mathcal{G}_i$ is a triplet that specifies the spatial goals and dynamic pattern,

$$\mathcal{G}_i = \{\mathcal{T}_H, \mathcal{T}_O, \mathcal{B}\}. \tag{4}$$

## 3.2 Automatic Policy Learning via VLM-Guided RMD

Upon receiving the plan generated by the VLM-guided RMD Planner, the policy produces joint-level torques to actuate the agent and execute the plan. Our physics-based HOI policy learning is formulated within the framework of goal-conditioned reinforcement learning. At each timestep $t$, the agent samples an action from its policy $\pi(a_t \mid \mathbf{s}_t, \mathbf{g}_t)$, conditioned on the current state $\mathbf{s}_t$ and the goal state $\mathbf{g}_t$. After executing the action, the environment transitions to the next state $\mathbf{s}_{t+1}$, and the agent receives two types of rewards: a task reward $r^G(\mathbf{s}_t, \mathbf{g}_t, \mathbf{s}_{t+1})$, which incentivizes behaviors aligned with achieving the specified goal $\mathbf{g}_t$, and a style reward $r^S(\mathbf{s}_t, \mathbf{s}_{t+1})$, which encourages the agent to produce natural behaviors as proposed in Peng et al. (2021). Unlike AMP and most prior methods that rely on manually specifying task-specific goal states and handcrafted reward terms, our framework supports the automatic construction of goal states and reward functions. In the following, we detail how our framework automatically generates both goal states and reward functions from VLM output, enabling the agent to learn behaviors aligned with the intended objectives.

**Automatic Goal State Construction via VLM-Guided RMD.** To provide the policy with fine-grained spatio-temporal guidance during the interaction, we encode the $i$-th planned interaction step $\mathcal{G}_i = \{\mathcal{T}_H, \mathcal{B}, \mathcal{T}_O\}$, generated by the VLM, with both kinematic and dynamic information.

For each edge $e_{ij} \in E$, we extract from the simulator the absolute position–velocity pair of the human joint, denoted by $\mathbf{p}_{h_i}^p, \mathbf{p}_{h_i}^v$, and that of the nearest surface point on the object part, $\mathbf{p}_{o_j}^p, \mathbf{p}_{o_j}^v$. We then compute their relative quantities in the agent-centric coordinate frame,

$$\tilde{\mathbf{p}}_{ij} = \mathbf{p}_{o_j}^p - \mathbf{p}_{h_i}^p, \quad \tilde{\mathbf{v}}_{ij} = \mathbf{p}_{o_j}^v - \mathbf{p}_{h_i}^v. \tag{5}$$

The edge weight $w_{ij} \in \{0, 1, 2, 3\}$ is encoded as a one-hot vector $\mathbf{w}_{ij}' \in \{0, 1\}^4$, and concatenated with the relative position and velocity to form the feature tuple $(\tilde{\mathbf{p}}_{ij}, \tilde{\mathbf{v}}_{ij}, \mathbf{w}_{ij}')$ at timestep $t$. Stacking these features across all edges yields the RMD state,

$$\mathbf{s}_t^{\text{RMD}} = \text{concat}_{(i,j) \in E} \left[ \tilde{\mathbf{p}}_{ij}, \tilde{\mathbf{v}}_{ij}, \mathbf{w}_{ij}' \right] \in \mathbb{R}^{|E| \times (3+3+4)}. \tag{6}$$

Both $\mathcal{T}_H$ and $\mathcal{T}_O$ generated by VLM are in the form `object(spatial-relationship)`, e.g., `armchair(front)`. For every referenced object we approximate its geometry by an axis-aligned bounding box with edge lengths $l_x, l_y, l_z$ and center $\mathbf{c}_{\text{obj}}$. We map each spatial-relationship token $\delta$ to a displacement vector $\Delta \mathbf{q}(\delta)$ in the object's local frame. For instance, `front` $\mapsto (0.7\, l_x, 0, 0)$. The absolute targets for the human root and the object root are defined as,

$$\mathbf{p}_{\text{tar}}^h = \mathbf{c}_{\text{obj}} + \Delta \mathbf{q}(\delta_h), \qquad \mathbf{p}_{\text{tar}}^o = \mathbf{c}_{\text{obj}} + \Delta \mathbf{q}(\delta_o). \tag{7}$$

The destinations proposed by the VLM, originally in the global coordinate system, are transformed into the agent-centric frame to produce the final destination state,

$$\mathbf{d}_t = \{\mathbf{d}_t^h, \mathbf{d}_t^o\}. \tag{8}$$

To improve local perception and obstacle avoidance, we attach a fixed-resolution heightmap $\mathbf{h}_t \in \mathbb{R}^{9 \times 9}$ that records the elevation of nearby geometry in the root frame (Tessler et al., 2024; Xiao et al., 2024). In parallel, we describe object dynamics by the axis-aligned bounding box vertices $\mathbf{V}_t^{\text{box}} \in \mathbb{R}^{8 \times 3}$, the rotation $\boldsymbol{\theta}_t \in \mathbb{R}^3$, the linear velocity $\mathbf{v}_t \in \mathbb{R}^3$, and the angular velocity $\boldsymbol{\omega}_t \in \mathbb{R}^3$,

$$\mathbf{o}_t = \left\{ \mathbf{V}_t^{\text{box}}, \ \boldsymbol{\theta}_t, \ \mathbf{v}_t, \ \boldsymbol{\omega}_t \right\}. \tag{9}$$

By concatenating the RMD state $\mathbf{s}_t^{\text{RMD}}$, the destination $\mathbf{d}_t$, the heightmap $\mathbf{h}_t$, and the object state $\mathbf{o}_t$, we construct the complete goal representation,

$$\mathbf{g}_t = \left\{ \mathbf{s}_t^{\text{RMD}}, \mathbf{d}_t, \mathbf{h}_t, \mathbf{o}_t \right\}. \tag{10}$$

**Automatic Reward Design via VLM-Guided RMD.** To guide human-object interaction in alignment with the $i$-th planned interaction step $\mathcal{G}_i = \{\mathcal{T}_H, \mathcal{B}, \mathcal{T}_O\}$ produced by the VLM, we design a composite reward function that promotes three key objectives: (i) guiding the human root trajectory towards the spatial target $\mathcal{T}_H$, (ii) guiding the object root trajectory towards $\mathcal{T}_O$, and (iii) enforcing the relative motion patterns prescribed by the RMD graph $\mathcal{B}$.

To realize these objectives, we define three corresponding reward terms. The term $r_d^{\text{h}}$ encourages the human root position $x_t^{\text{h}}$ to approach the designated target $\mathbf{d}_t^h$, and $r_d^{\text{o}}$ encourages the object root position $x_t^{\text{o}}$ to move toward its designated target $\mathbf{d}_t^o$,

$$r_d^{\text{h}} = \exp\left(-\left\|x_t^{\text{h}} - \mathbf{d}_t^h\right\|^2\right), \qquad r_d^{\text{o}} = \exp\left(-\left\|x_t^{\text{o}} - \mathbf{d}_t^o\right\|^2\right). \tag{11}$$

The term $r_{\text{RMD}}$ encourages each edge $e_{ij} \in \mathcal{B}$, representing a *(human-part, object-part)* pair, to follow the relative motion dynamics specified by its associated edge weight $w_{ij}$, as planned by the VLM. Specifically, $r_{\text{RMD}}$ is computed by summing the individual alignment reward $r_{\text{rmd}}^{ij}(\cdot)$, each of which measures how well the pair $(p_{h_i}, p_{o_j})$ follows the motion pattern defined by $w_{ij}$, weighted by a coefficient $\lambda_{ij}$,

$$r_{\text{RMD}} = \sum_{(i,j) \in E} \lambda_{ij} \cdot r_{\text{rmd}}(\tilde{\mathbf{p}}_{ij}, \tilde{\mathbf{v}}_{ij}, w_{ij}). \tag{12}$$

Further implementation details of $r_{\text{rmd}}^{ij}(\cdot)$ are provided in the appendix. Combining these terms, the overall task reward at time $t$ is given by,

$$r^G\left(\mathbf{s}_t, \mathbf{g}_t, \mathbf{s}_{t+1}\right) = \lambda_{\text{RMD}} \cdot r_{\text{RMD}} + \lambda_h \cdot r_d^{\text{h}} + \lambda_o \cdot r_d^{\text{o}}. \tag{13}$$

Once the task reward $r^G$, which is bounded within the range $[0, 1]$, exceeds 0.9, the policy transitions to the next planned interaction step $\mathcal{G}_{i+1}$ at the following timestep. To balance the contributions of different reward terms, we adopt adaptive weighting strategies as proposed by Xiao et al. (2024), which dynamically adjust the value of $\lambda$ in Eq. 12 and Eq. 13. This design enables the current reward to be evaluated in a balanced manner and facilitates accurate assessment of progress within the current stage, thereby ensuring the appropriate timing for transitioning to the next goal.

The style reward $r^S(\mathbf{s}_t, \mathbf{s}_{t+1})$ is computed by a discriminator $D(\mathbf{s}_{t-10:t})$ based on a window of 10 consecutive states. Combining the task reward $r^G$ and the style reward $r^S$, the total reward at timestep $t$ is defined as,

$$r_t = \alpha_{\text{task}}\, r^G\left(\mathbf{s}_t, \mathbf{g}_t, \mathbf{s}_{t+1}\right) + \alpha_{\text{style}}\, r^S\left(\mathbf{s}_t, \mathbf{s}_{t+1}\right). \tag{14}$$

Table 2: Comparison with baselines in a long-horizon multi-task scenario.

| Methods | Completion Rate (%) ↑ | | | Sub-step Completion Ratio (%) ↑ | | | Sub-step Precision (cm) ↓ | | |
|---|---|---|---|---|---|---|---|---|---|
| | Static Interaction | Dynamic Interaction | Hybrid | Static Interaction | Dynamic Interaction | Hybrid | Static Interaction | Dynamic Interaction | Hybrid |
| InterPhys* Hassan et al. (2023) | 21.3 | 47.8 | 27.5 | 37.3 | 61.9 | 54.1 | 13.8 | 18.7 | 16.9 |
| TokenHSI* Pan et al. (2025) | 25.2 | 52.5 | 36.0 | 48.1 | 65.7 | 60.1 | 13.1 | 16.6 | 14.4 |
| UniHSI Xiao et al. (2024) | 37.2 | - | - | 61.3 | - | - | 10.2 | - | - |
| **Ours** | **75.1** | **71.2** | **53.8** | **86.2** | **84.3** | **71.8** | **7.7** | **13.0** | **11.2** |
| Ours w/ LLM | 62.8 | 53.1 | 39.9 | 81.7 | 78.3 | 67.2 | 8.9 | 15.2 | 13.8 |

## 4 EXPERIMENT

We structure our experiments into two main parts: evaluating HOI skill learning in a complex long-horizon multi-task scenario (Sec. 4.1) and in a simple single-task scenario (Sec. 4.2).

**InterPlay Dataset.** To evaluate our framework in a long-horizon multi-task scenario, we construct a novel dataset that includes both static and dynamic interaction tasks under diverse indoor scene contexts. Each sample comprises four components: (i) a set of processed 3D assets that can be directly deployed in simulation, (ii) the corresponding 3D scene layout, (iii) a top-view rendered image, and (iv) a natural-language instruction specifying the target task. We curate high-quality 3D object assets from established resources, including PartNet (Mo et al., 2019), 3D-FRONT (Fu et al., 2021), SAMP (Hassan et al., 2021), CIRCLE (Araújo et al., 2023), and PartNet-Mobility (Xiang et al., 2020). Each scene contains at least two interactive objects and no fewer than two furniture assets, providing richer contextual diversity and increased HOI complexity. Additional details of the dataset construction are provided in the appendix.

**Implementation Details.** To ensure our agent interacts with objects in a natural way, we collect motion clips from SAMP (Hassan et al., 2023), OMOMO (Li et al., 2023a), CIRCLE (Araújo et al., 2023). We conduct experiments in parallel simulated environments within Isaac Gym (Makoviychuk et al., 2021), using PyTorch for neural network implementation. In line with previous studies (Hassan et al., 2023; Xiao et al., 2024), our physical humanoid model consists of 15 rigid body parts and 28 joints, all controlled by a PD controller. We utilize Proximal Policy Optimization (PPO) (Schulman et al., 2017) to train the policy network on an NVIDIA RTX 3090 GPU.

### 4.1 EVALUATION IN A LONG-HORIZON MULTI-TASK SCENARIO

Real-world applications require agents to perform multiple tasks concurrently or sequentially, requiring advanced coordination and planning. In this context, the agent must interact with multiple objects in sequence according to the plan, which poses significant challenges for ensuring the robustness of VLM-guided motion policy learning and for managing long-horizon transitions.

**Settings.** We split our dataset into three categories: static-interaction, dynamic-interaction, and hybrid setups. In the static-only setup, we sequentially interact with at least two static objects as planned in the dataset. In the dynamic-only setup, we interact with at least two dynamic objects or articulated objects. In the hybrid setup, we select at least three objects, ensuring that at least one dynamic human-object interaction task is included in the overall plan. To ensure fair comparisons under this taxonomy, we align each baseline with the subset of settings it can natively support. UniHSI (Xiao et al., 2024) supports only sequential interactions with static objects. We adapt its released prompt to generate plans based on scenarios from our static-only dataset. Both Inter-Phys (Hassan et al., 2023) and TokenHSI (Pan et al., 2025) support interactions with both static and dynamic objects, but they only cover a subset of HOI tasks. For methods without native multi-task planning, we introduce a standardized adaptation protocol before evaluation. To enable multi-task evaluation, we re-implement their methods by training individual skills in parallel across different environments, manually designing goal states and corresponding reward functions for each. A rule-based finite-state machine is employed to switch between tasks during execution. This adaptation requires substantial manual effort.

**Metrics.** We report the **Completion Rate**, which measures the percentage of trials in which the character sequentially completes all sub-step tasks and returns to a natural standing pose, with the

Table 3: Comparison with baselines in a single-task scenario.

| Methods | Completion Rate (%) ↑ | | | | | | Success Rate (%) ↑ | | | | | | Precision (cm) ↓ | | | | | |
|---|---|---|---|---|---|---|---|---|---|---|---|---|---|---|---|---|---|---|
| | Carry | Push | Open | Sit | Lie | Reach | Carry | Push | Open | Sit | Lie | Reach | Carry | Push | Open | Sit | Lie | Reach |
| AMP* Peng et al. (2021) | 53.2 | 40.4 | 63.2 | 7.4 | 0.9 | 93.2 | 74.9 | 71.0 | 70.3 | 77.4 | 51.9 | 92.2 | 17.3 | 31.7 | 18.1 | 13.2 | 19.1 | 5.5 |
| InterPhys* Hassan et al. (2023) | 67.8 | 47.1 | 83.2 | 23.2 | 2.7 | 95.3 | 88.2 | 78.1 | 90.1 | 88.1 | 76.4 | 95.9 | 12.3 | 24.7 | 10.1 | 10.3 | 14.8 | 3.4 |
| TokenHSI* Pan et al. (2025) | 71.2 | 49.3 | 81.1 | 27.8 | 8.9 | 95.7 | 92.2 | 79.5 | 90.8 | 92.8 | 78.2 | 95.9 | 11.2 | 22.5 | 8.7 | 9.8 | 13.2 | 3.5 |
| UniHSI Xiao et al. (2024) | - | - | - | 58.9 | 23.2 | 97.1 | - | - | - | 94.3 | 81.5 | 97.5 | - | - | - | 5.6 | 12.8 | 2.1 |
| Ours | **88.3** | **84.1** | **91.2** | **92.6** | **62.0** | **97.5** | **93.3** | **90.1** | **95.1** | **95.6** | **87.0** | **97.8** | **10.1** | **18.2** | **5.2** | **4.9** | **9.1** | **1.9** |
| multi-one | 75.1 | 71.3 | 77.9 | 81.0 | 49.2 | 96.2 | 90.5 | 86.4 | 92.1 | 94.2 | 83.1 | 97.2 | 13.6 | 21.0 | 7.3 | 5.5 | 14.3 | 2.0 |
| one-one | 11.7 | 59.2 | 68.7 | 65.8 | 28.1 | 97.2 | 14.3 | 81.2 | 90.8 | 92.3 | 81.6 | 97.4 | 21.3 | 32.2 | 7.2 | 9.1 | 17.2 | 2.0 |
| w.o. $\tilde{\mathbf{p}}_{ij}$ | 71.7 | 73.8 | 77.0 | 72.9 | 36.8 | 96.8 | 88.9 | 87.8 | 90.3 | 91.8 | 79.8 | 96.8 | 10.9 | 19.7 | 10.1 | 5.6 | 12.8 | 2.9 |
| w.o. $\tilde{\mathbf{v}}_{ij}$ | 78.2 | 76.7 | 75.2 | 76.1 | 42.1 | 96.9 | 90.1 | 84.2 | 87.8 | 92.1 | 79.2 | 96.9 | 11.3 | 20.0 | 9.9 | 6.1 | 14.2 | 2.5 |
| w.o. $w_{ij}^{f}$ | 69.1 | 68.4 | 62.1 | 67.0 | 30.3 | 95.1 | 82.3 | 82.3 | 86.2 | 90.3 | 78.9 | 95.2 | 14.1 | 23.9 | 9.2 | 7.6 | 18.6 | 2.2 |

character's root within 20 cm of the target position. We introduce the **Sub-step Completion Ratio**, which measures the ratio of completed sub-steps to the total number of planned sub-steps. This metric provides a more granular view of the agent's ability to handle complex tasks with multiple sub-goals. Additionally, we use **Sub-step Precision** to measure the average distance between a specific human part and its target, or between the object root and its target.

**Result Analysis.** As shown in Tab. 2, our approach achieves state-of-the-art performance in both individual sub-step execution and long-horizon transitions, reflecting its robustness in handling intricate sequences of actions. Our method benefits from a unified formulation of interaction and fine-grained spatio-temporal guidance for all human body parts, enabling seamless transitions to a multi-task setup. This design not only streamlines the training process by maintaining consistent representations across various tasks but also significantly improves the model's ability to generalize across different interaction scenarios. In the absence of a unified formulation, approaches that naively combine multiple tasks, such as InterPhys and TokenHSI, often encounter difficulties in smoothly switching between tasks. In contrast, our integrated method preserves critical spatial and temporal information throughout the pipeline. While UniHSI excels at approaching and interacting with static objects, it struggles with actions such as getting up after interaction, which notably limits its performance in long-horizon scenarios.

**Ablation.** While both VLMs and LLMs possess extensive world knowledge, VLMs exhibit superior spatial awareness and motion imagination thanks to their integration of visual and textual inputs. To evaluate this, we replaced top-view images with textual descriptions and assessed an LLM-based planner under purely textual conditions. The resulting performance drop highlights the essential role of VLMs in our framework. Pretrained on diverse vision-language data, the VLM excels at generating semantically meaningful and spatially grounded plans — capabilities critical for skill learning in complex, multi-object environments. Without VLM guidance, the system struggles to produce coherent long-horizon behaviors, leading to lower task success and degraded motion quality.

## 4.2 EVALUATION IN A SINGLE-TASK SCENARIO

**Settings.** We select a set of interaction tasks from our InterPlay dataset, including both tasks commonly used in prior work and several novel tasks not previously explored. These include three static HOI tasks: reaching, sitting, and lying down, and three dynamic HOI tasks: carrying, pushing, and opening. This setup lets us evaluate not only whether an interaction is achieved, but also whether agents can transition reliably across heterogeneous tasks. While previous works (Hassan et al., 2023; Xiao et al., 2024) primarily focus on approaching and interacting with objects, they often overlook a critical aspect: after interacting, the agent must return to a neutral state to facilitate subsequent interactions. For instance, in the sitting task, previous methods consider the task complete once the agent is seated. However, this disregards the need for the agent to return to standing in order to transition smoothly to the next task. Therefore, we redefine the task completion criteria by introducing a "leaving" step: after interacting with an object, the agent must stand up and walk to a designated position. This step ensures that the agent returns to a neutral pose, which is essential for smooth transitions between tasks. For each episode, objects are initialized with a random orientation (ranging from 0 to $2\pi$), a random distance (from 4 to 10 meters), and a random scale (from 0.8 to 1.2). The finish position is then randomly sampled at a point 3 meters away from the object. To ensure a fair comparison with previous state-of-the-art methods, we rigorously followed their estab-

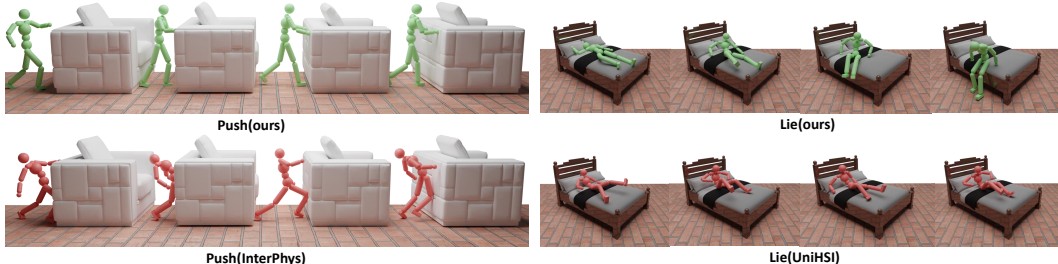

Figure 3: **Visualization for qualitative comparison.** Other methods exhibit unnatural motion (InterPhys) or incomplete interactions (UniHSI), whereas our method demonstrates human-like motion quality in qualitative assessments. More qualitative visualization videos can be found in the supplementary materials.

lished implementation protocols and re-implemented AMP (Peng et al., 2021), InterPhys (Hassan et al., 2023), and TokenHSI (Pan et al., 2025), modifying goal states and reward functions only as necessary to adapt them to our experimental setup.

**Metrics.** We first introduce a new metric, **Completion Rate**, to evaluate the percentage of trials in which the character not only completes the interaction with the object but also returns to a neutral standing pose, with the root located within 20 cm of the designated finish position. This metric is designed to reflect the agent's ability to recover from interaction and complete the full motion sequence, offering a more comprehensive evaluation of control quality. In line with prior works (Hassan et al., 2023; Xiao et al., 2024), we also report the **Success Rate**, which measures the percentage of trials in which the character successfully interacts with the object. A trial is considered successful if the distance between the object's root and a specific humanoid part is less than 20 cm (for tasks like reaching, sitting, or lying down), or if the distance between the object's root and the target location is less than 20 cm (for tasks like carrying, pushing, or opening). Furthermore, we report **Precision** to quantify the average distance between a specific human body part and its corresponding target position over the course of the entire interaction in successfully completed trials. All metrics are evaluated over 4096 trials.

**Result Analysis.** As shown in Tab. 3, our method achieves higher or comparable performance across all these metrics compared to the baselines. Our method outperforms tasks involving the recovery process of static human-object interactions, such as getting up after sitting or lying down, as the Relative Movement Dynamics guides each body part to move away from the object. In contrast, as shown in Fig. 3, UniHSI (Xiao et al., 2024) struggles to get up because it treats the human-object interaction as a sequence of independent spatial reaching tasks, neglecting the temporal dynamics that coordinate the movements of different body parts. Similarly, InterPhys (Hassan et al., 2023) produces unnatural and unstable motions to complete tasks, such as abrupt kicking or thrusting forward, due to its lack of fine-grained spatio-temporal guidance for coordinating the movements of different body parts. Instead, our design ensures that the agent keeps both hands and arms relatively stationary against the back of the sofa.

**Ablation.** We conduct an ablation study on key components to evaluate the contribution of the proposed VLM-guided RMD. In the multi-one setup, we simplify object representation by removing the division into distinct parts and treating the object as a single entity. Representing the object only by its root pose and kinematic data leads to a slight performance drop, as the model loses fine-grained geometric information necessary for accurate interaction modeling. In the one-one setup, we restrict the VLM-guided RMD Planner to model only the dynamics of a single human part interacting with one object part per sub-step. This disrupts whole-body coordination and often traps optimization in local optima, resulting in a noticeable decline in performance. These results highlight the effectiveness of the multi-multi formulation in handling complex interactions. Moreover, removing either kinematic relation encoding ($\tilde{p}_t^{ij}$, $\tilde{v}_t^{ij}$) or dynamic encoding ($w_t'^{ij}$) further degrades performance, as both are crucial for capturing the spatio-temporal relationships between human and object. Overall, the ablation results confirm the central role of VLM-guided RMD in our framework.

## 5 CONCLUSION

In this paper, we present a physics-based framework for synthesizing diverse Human-Object Interactions, guided by VLMs, through our proposed Relative Movement Dynamics — a spatio-temporal representation that structures part-level relationships between human and object. RMD automates the generation of goal states and reward functions for reinforcement learning, eliminating manual engineering while supporting static, dynamic, and articulated interactions. Our method bridges high-level semantic reasoning with low-level physics-based control, producing natural motions across diverse interactions that outperform task-specific and LLM-based baselines in realism and adaptability. Beyond performance gains, RMD offers a unified representation that may be useful for extending to broader HOI settings. To facilitate evaluation, we introduce the Interplay dataset for long-horizon HOI tasks. Extensive results demonstrate the effectiveness of our approach on both single-task and long-horizon multi-task scenarios.

## 6 LIMITATIONS, EXTENSIONS, AND FUTURE WORK

**Limitations.** While our framework advances dynamic human-object interaction modeling, it focuses on single-agent scenarios and does not address multi-agent interactions (e.g. collaborative tasks involving multiple agents) or social dynamics (e.g. social impact considerations in shared environments). This restricts its applicability to real-world settings requiring coordination among multiple agents, such as collaborative kitchens or assistive care scenarios. Additionally, the current VLM-based planner may struggle with long-horizon tasks that demand temporal reasoning and hierarchical planning. For instance, tasks like cooking multi-step meals or tidying interconnected household spaces remain challenging due to the lack of explicit task decomposition mechanisms.

**Extending to Real-World Applications.** We view RMD as a mid-level relational abstraction that bridges semantic intent and executable control by explicitly modeling the spatio-temporal relationships among interacting rigid bodies. This graph formulation is naturally extensible: by enriching the vertex and edge sets, RMD can encode additional entities and constraints (e.g., multiple agents, articulated objects, tool use, and self-interaction), while preserving the same downstream policy-learning interface. In simulation, privileged part-level human/object states are available, allowing our framework to automatically propose goal states and construct rewards for policy training, yielding strong controllers without manual engineering. During deployment in real-world or partially observed settings where these signals are inaccurate, the trained controllers can serve as teachers in a distillation/imitation pipeline from egocentric sensory inputs, followed by lightweight adaptation (e.g., residual RL or teacher-student distillation). Finally, robustness can be improved by strengthening knowledge grounding and multi-step reasoning (e.g., retrieval-augmented planning and structured deliberation) and by adding a dedicated verification stage that filters or repairs infeasible plans before policy execution.

**Future Work.** Beyond these directions, we will explore integrating diffusion-based generative priors to synthesize more diverse and natural interaction motions, which may further improve coverage of rare but realistic behaviors. We are also interested in joint training paradigms that more tightly couple the VLM planner and the physics-based controller (e.g., via alternating optimization or planner–controller consistency objectives), with the goal of reducing planner–controller mismatch and improving both generalization and control fidelity in complex, long-horizon interactions. In addition, we plan to extend policy learning to multi-agent settings with learned coordination mechanisms, such as adaptive communication and dynamic role allocation, to support collaborative household activities at scale.

## ACKNOWLEDGEMENTS

This work was supported by National Natural Science Foundation of China (62406195, 62303319), Shanghai Local College Capacity Building Program (23010503100), ShanghaiTech AI4S Initiative SHTAI4S202404, HPC Platform of ShanghaiTech University, Core Facility Platform of Computer Science and Communication of ShanghaiTech University, and MoE Key Laboratory of Intelligent Perception and Human-Machine Collaboration (ShanghaiTech University), and Shanghai Engineering Research Center of Intelligent Vision and Imaging.

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

APPENDICES

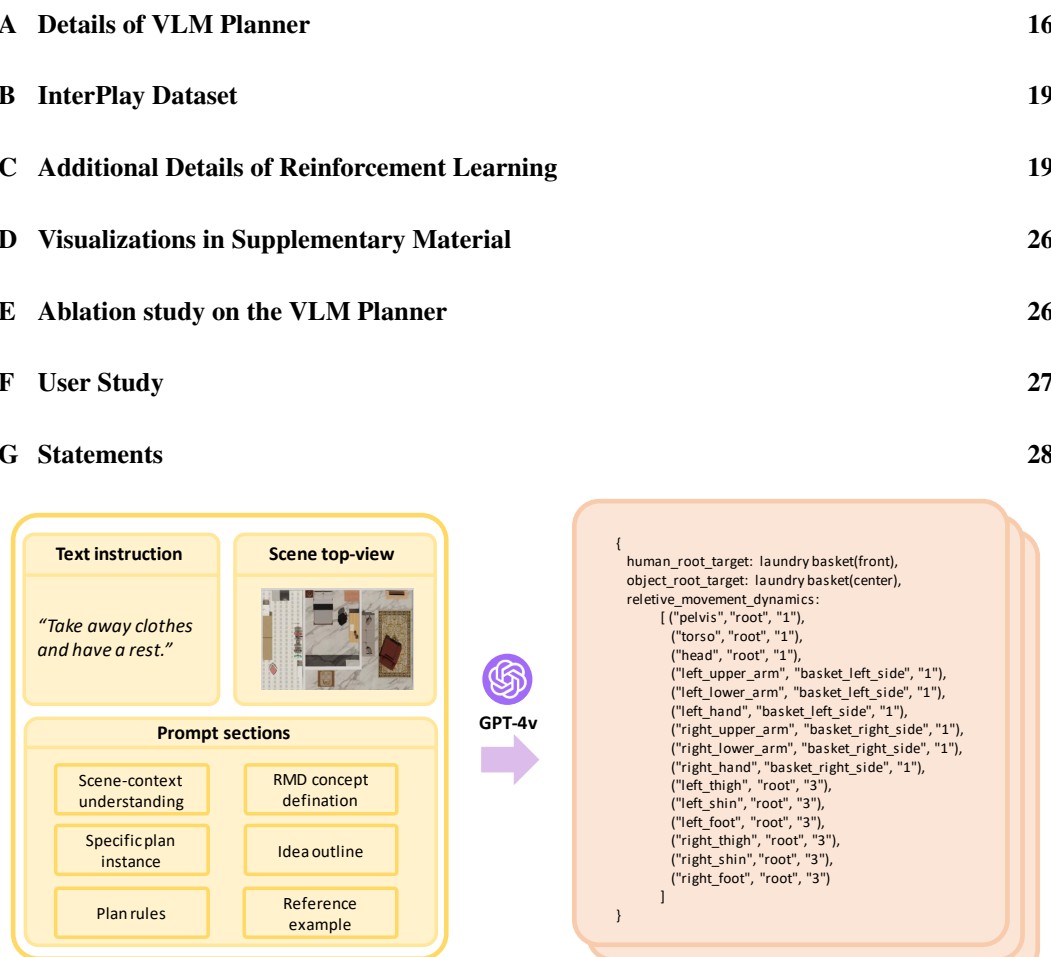

Figure 4: An overview of VLM-guided RMD Planner pipeline.

## A    DETAILS OF VLM PLANNER

As shown in Fig. 4, given a top-view image of the surrounding scene and a textual instruction, the VLM-guided RMD Planner generates a sequence of sub-step plans in the form of structured JSON. This output format facilitates direct downstream processing via Python scripts.

To ensure that the planner performs as intended, we design prompts that explicitly guide it to understand the concept of Relative Movement Dynamics (RMD), perceive the contextual scene information, decompose high-level instructions, and produce outputs in a structured format consistent with our framework. For clarity and modularity, we organize the prompt into distinct sections, each fulfilling a specific functional role as illustrated in Fig. 5 and Fig. 6. The content of each section is detailed below.

**Scene-context Understanding.** This section of the prompt helps the planner interpret the scene context and determine the target destination using a simplified relative spatial relationship. Specifically, we reduce spatial relations to seven canonical types: *center*, *forward*, *back*, *left*, *right*, *up*, and *down*.

## Prompt section1: scene-context understanding

To effectively plan sequential interactions between humanoid and objects within a scene, it is essential to recognize the type of each object, understand the spatial relationships among them, and identify the most suitable object for interaction at each sub-step, based on the scene's top-down view. We simplify the spatial relationships between objects into seven types: center, forward, back, left, right, up, and down. For example, you can use the term "box (front)" to describe the position that lies in front of the box.

## Prompt section3: specific plan instance

```
{
    human_root_target: box(front),
    object_root_target: box(center),
    reletive_movement_dynamics:
        [
            ("pelvis", "root", "1"),
            ("torso", "root", "1"),
            ("head", "root", "1"),
            ("left_upper_arm", "box_left_side", "0"),
            ("left_lower_arm", "box_left_side", "0"),
            ("left_hand", "box_left_side", "0"),
            ("right_upper_arm", "box_right_side", "0"),
            ("right_lower_arm", "box_right_side", "0"),
            ("right_hand", "box_right_side", "0"),
            ("left_thigh", "root", "3"),
            ("left_shin", "box_left_side", "2"),
            ("left_foot", "box_left_side", "2"),
            ("right_thigh", "root", "3"),
            ("right_shin", "box_right_side", "2"),
            ("right_foot", "box_right_side", "2")
        ]
}
```
This instance describes the relative movement dynamics during a scenario where a humanoid bends down to grab a box with both hands and picks it up.

- The "pelvis", "torso", and "head" have an edge weight of 1 with "root" because the distance between the pelvis, torso, head and the box center decreases as the box is gradually lifted.

- The "left_upper_arm", "left_lower_arm", and "left_hand" have an edge weight of 0 with "box_left_side", indicating that these body parts remain relatively stationary with respect to the box side during this time.

- The "left_shin", "left_foot" and "box_left_side" have an edge weight of 3 as the box is leaving with lower part of body.

- The rest of the weight can be obtained in the same way.

- The humanoid does not need to move during interaction, therefore the human_root_target correspond to the position in the front of the box.

- The object does not need to be moved in xy plane during the interaction, therefore the object_root_target corresponds to the position at the center of the box.

## Prompt section4: idea outline

Complex human-object interaction (HOI) tasks can be decomposed into several sub-sequences, where each human part maintains a consistent relative movement trend with its corresponding object part until the transition to the next sub-sequence. Therefore, we universally define the objective of the interaction plan as D, with the formulation,
$$D = \{G_1, G_2, ..., G_N\}$$
where $G_i$ denotes the i-th plan interaction step. I will provide you with an instruction and a top-view image depicting the surrounding scene context. Your task is to break down the instruction into sub-sequences represented as relative movement dynamics, based on the top-view image. Please note that for different HOI tasks, the granularity of an object's part decomposition may vary. This granularity is determined by the scene context and the instruction itself.

## Prompt section2: RMD concept definition

The **Relative Movement Dynamics** represents the relative movement trend between each human body part and the corresponding object part over a period of time. This trend is modeled as a bipartite graph, where one subset of nodes represents human body parts, and the other subset represents parts of an object.

The subset representing human body parts consists of fifteen nodes corresponding to specific regions of the body:
- pelvis           - torso            - head
- right_upper_arm  - right_lower_arm  - right_hand
- left_upper_arm   - left_lower_arm   - left_hand
- right_thigh      - right_shin       - right foot
- left_thigh       - left_shin        - left_foot

The subset representing the object parts contains $x$ nodes, where $x$ depends on the specific object and instruction. You are expected to infer a reasonable partitioning of the object into parts. Note that your partition should reflect common divisions of objects as they are understood in everyday life.

i.e. (human_body_part, object_body_part, edge_weight)

Each edge in this graph can take on a weight of 0, 1, 2, or 3, where:
- 0 indicates that the body part and object part connected by this edge remain relatively stationary with respect to each other over time.
- 1 indicates that the distance between the body part and the object part connected by this edge is steadily decreasing over time.
- 2 indicates that the distance between the body part and the object part connected by this edge is steadily increasing over time.
- 3 indicates that the relative movement trend between the human part and the object part is unstable or unclear over time.

## Prompt section5: plan rules

1. All instructions pertain to the interaction task between a humanoid robot, modeled with relative simplicity and equipped with a ball rather than a finger on its hand, and an object. This interaction may be dynamic or static.

2. Each plan may involve sequential interactions with multiple objects. For each object, you should design a plan comprising at least five steps, including approaching and departing from the object.

3. Your return about relative_movement_dynamics should be formatted into sequence of [(human_body_part, object_body_part, edge_weight), (human_body_part, object_body_part, edge_weight), ...], 15 edges for one sequence.

4. If you are confused about which part of the object you should select, you can select the root to represent the whole.

5. Be relatively sensitive to the judgment of relative movement dynamics, if it is not stable(e.g. when holding a box and walking towards destination, the feet and the box will alternately approach and move away from each other for a period of time), just set the weight to 3.

6. Your thought process may unfold as follows: perceive the surrounding scene context and identify the relevant object, decompose the instruction into a sequence of human–object interactions, imagine the corresponding motions, and represent each sub-sequence in the form of RMD.

Figure 5: Details of prompt section.

**Prompt section6: reference example**

Instruction: Take away clothes and have a rest.

Top-view image:

```
step1: get close to the laundry basket.
{
    human_root_target: laundry basket(back),
    object_root_target: laundry basket(center),
    reletive_movement_dynamics:
        [
            ("pelvis", "root", "1"),
            ("torso", "root", "1"),
            ("head", "root", "1"),
            ("left_upper_arm", "basket_left_side", "1"),
            ("left_lower_arm", "basket_left_side", "1"),
            ("left_hand", "basket_left_side", "1"),
            ("right_upper_arm", "basket_right_side", "1"),
            ("right_lower_arm", "basket_right_side", "1"),
            ("right_hand", "basket_right_side", "1"),
            ("left_thigh", "root", "3"),
            ("left_shin", "root", "3"),
            ("left_foot", "root", "3"),
            ("right_thigh", "root", "3"),
            ("right_shin", "root", "3"),
            ("right_foot", "root", "3")
        ]
}
```

**Prompt section6: reference example**

```
step2: bends down to grab the box with both hands and
picks it up.
{
    human_root_target: laundry basket(back),
    object_root_target: laundry basket(center),
    reletive_movement_dynamics:
        [
            ("pelvis", "root", "1"),
            ("torso", "root", "1"),
            ("head", "root", "1"),
            ("left_upper_arm", "basket_left_side", "0"),
            ("left_lower_arm", "basket_left_side", "0"),
            ("left_hand", "basket_left_side", "0"),
            ("right_upper_arm", "basket_right_side", "0"),
            ("right_lower_arm", "basket_right_side", "0"),
            ("right_hand", "basket_right_side", "0"),
            ("left_thigh", "root", "3"),
            ("left_shin", "basket_left_side", "2"),
            ("left_foot", "basket_left_side", "2"),
            ("right_thigh", "root", "3"),
            ("right_shin", "basket_right_side", "2"),
            ("right_foot", "basket_right_side", "2")
        ]
}
```

```
step3: walking to the washing machine while holding the
laundry basket.
{
    human_root_target: washing machine(front),
    object_root_target: washing machine(front),
    reletive_movement_dynamics:
        [
            ("pelvis", "root", "0"),
            ("torso", "root", "0"),
            ("head", "root", "0"),
            ("left_upper_arm", "basket_left_side", "0"),
            ("left_lower_arm", "basket_left_side", "0"),
            ("left_hand", "basket_left_side", "0"),
            ("right_upper_arm", "basket_right_side", "0"),
            ("right_lower_arm", "basket_right_side", "0"),
            ("right_hand", "basket_right_side", "0"),
            ("left_thigh", "root", "3"),
            ("left_shin", "root", "3"),
            ("left_foot", "root", "3"),
            ("right_thigh", "root", "3"),
            ("right_shin", "root", "3"),
            ("right_foot", "root", "3")
        ]
}
```

**Prompt section6: reference example**

```
step4: put down the laundry basket.
{
    human_root_target: washing machine(front),
    object_root_target: washing machine(front),
    reletive_movement_dynamics:
        [
            ("pelvis", "root", "2"),
            ("torso", "root", "2"),
            ("head", "root", "2"),
            ("left_upper_arm", "basket_left_side", "0"),
            ("left_lower_arm", "basket_left_side", "0"),
            ("left_hand", "basket_left_side", "0"),
            ("right_upper_arm", "basket_right_side", "0"),
            ("right_lower_arm", "basket_right_side", "0"),
            ("right_hand", "basket_right_side", "0"),
            ("left_thigh", "root", "3"),
            ("left_shin", "basket_left_side", "1"),
            ("left_foot", "basket_left_side", "1"),
            ("right_thigh", "root", "3"),
            ("right_shin", "basket_right_side", "1"),
            ("right_foot", "basket_right_side", "1")
        ]
}
```

```
step5: get close to the chair.
{
    human_root_target: chair(front),
    object_root_target: chair(center),
    reletive_movement_dynamics:
        [
            ("pelvis", "root", "1"),
            ("torso", "root", "1"),
            ("head", "root", "1"),
            ("left_upper_arm", "root", "1"),
            ("left_lower_arm", "root", "1"),
            ("left_hand", "root", "1"),
            ("right_upper_arm", "root", "1"),
            ("right_lower_arm", "root", "1"),
            ("right_hand", "root", "1"),
            ("left_thigh", "root", "3"),
            ("left_shin", "root", "3"),
            ("left_foot", "root", "3"),
            ("right_thigh", "root", "3"),
            ("right_shin", "root", "3"),
            ("right_foot", "root", "3")
        ]
}
```

**Prompt section6: reference example**

```
step6: sit down.
{
    human_root_target: chair(center),
    object_root_target: chair(center),
    reletive_movement_dynamics:
        [
            ("pelvis", "seat_support", "1"),
            ("torso", "root", "3"),
            ("head", "root", "3"),
            ("left_upper_arm", "root", "3"),
            ("left_lower_arm", "root", "3"),
            ("left_hand", "root", "3"),
            ("right_upper_arm", "root", "3"),
            ("right_lower_arm", "root", "3"),
            ("right_hand", "root", "3"),
            ("left_thigh", "chair_foot", "0"),
            ("left_shin", "chair_foot", "0"),
            ("left_foot", "chair_foot", "0"),
            ("right_thigh", "chair_foot", "0"),
            ("right_shin", "chair_foot", "0"),
            ("right_foot", "chair_foot", "0")
        ]
}
```

```
step7: have a rest.
{
    human_root_target: chair(center),
    object_root_target: chair(center),
    reletive_movement_dynamics:
        [
            ("pelvis", "seat_support", "0"),
            ("torso", "back_soft_support", "1"),
            ("head", "head_pillow", "1"),
            ("left_upper_arm", "root", "3"),
            ("left_lower_arm", "left_arm_sofa", "1"),
            ("left_hand", "root", "3"),
            ("right_upper_arm", "root", "3"),
            ("right_lower_arm", "right_arm_sofa", "1"),
            ("right_hand", "root", "3"),
            ("left_thigh", "chair_foot", "0"),
            ("left_shin", "chair_foot", "0"),
            ("left_foot", "chair_foot", "0"),
            ("right_thigh", "chair_foot", "0"),
            ("right_shin", "chair_foot", "0"),
            ("right_foot", "chair_foot", "0")
        ]
}
```

Figure 6: Details of prompt section.

These relative positions are defined with respect to a local coordinate system centered on the reference object.

**RMD Concept Defination.** This section of the prompt helps the planner understand the concept of RMD, clarifies its intended semantics, introduces a method for instantiating it, and provides a formal definition of its instantiated form.

**Specific Plan Instance.** This section presents a templated instance of an RMD-based plan, illustrating the correspondence between the template elements and RMD components, as well as how such an instance aligns with a real-world interaction scenario. This helps the planner internalize the concept and encourages outputs that conform to the desired format.

**Idea Outline.** This section offers a high-level overview of our approach, specifying the expected input and output of the interaction planner. Human–object interaction tasks are decomposed into a sequence of sub-steps, each characterized by consistent movement dynamics.

**Plan Rules.** This section enumerates the rules that the planner must follow, emphasizing their significance in prompt engineering. These rules define task interaction boundaries, output formatting, and criteria for determining movement dynamics. By adhering to these rules, the planner can effectively decompose complex interactions into manageable and coherent sub-sequences.

**Reference Example.** Empirical findings from the NLP community indicate that one-shot prompting often yields significantly better results than zero-shot prompting. Guided by this insight, we include a complete input–output example in the prompt to help the planner better understand the task expectations and output format.

## B  INTERPLAY DATASET

To evaluate our framework in long-horizon, multi-task scenarios, we construct a novel dataset, *InterPlay*, which encompasses both static and dynamic interaction tasks under diverse indoor scene contexts. Each data sample includes: (i) a list of processed 3D assets ready for direct use in simulation, (ii) the corresponding 3D scene layout, (iii) a top-view rendered image, and (iv) a natural language instruction describing the intended task.

In building the *InterPlay* dataset, we incorporate high-quality 3D object assets from several existing datasets, including PartNet (Mo et al., 2019), 3D-FRONT (Fu et al., 2021), SAMP (Hassan et al., 2021), CIRCLE (Araújo et al., 2023), and PartNet-Mobility (Xiang et al., 2020). From these sources, we carefully select 117 assets spanning categories such as bed, chair, sofa, washing machine, box, door, window, and wardrobe. Each asset is normalized, segmented into functional parts, and associated with part-level point clouds, based either on existing annotations or manual labeling.

To generate scene layouts, we apply a set of heuristic rules that ensure: (i) a well-distributed arrangement without object collisions, (ii) semantically meaningful placements aligned with typical household settings, and (iii) a minimum of two interactive objects and at least two furniture items per scene, thereby enriching the contextual complexity of human–object interaction. We then populate these layout templates by randomly sampling from the processed asset pool.

For each completed layout, we render a top-view image and employ GPT-4V Achiam et al. (2023) to generate a corresponding task instruction grounded in the scene context. Finally, we apply our VLM-guided RMD Planner to produce interaction plans, which are manually reviewed and refined before inclusion in the dataset.

In total, *InterPlay* contains 1,210 interaction plans, covering a wide range of HOI behaviors involving static, dynamic, and articulated objects with varying temporal lengths, all situated in realistic, household-inspired environments.

## C  ADDITIONAL DETAILS OF REINFORCEMENT LEARNING

Our physics-based animation framework is built upon goal-conditioned reinforcement learning. At each time step $t$, the agent samples an action from its policy $\pi(a_t \mid \mathbf{s}_t, \mathbf{g}_t)$ based on the current state $\mathbf{s}_t$ and the goal state $\mathbf{g}_t$. After executing the action, the environment transitions to the next state $\mathbf{s}_{t+1}$, and the agent receives a task reward $r^G(\mathbf{s}_t, \mathbf{g}_t, \mathbf{s}_{t+1})$. Further details are provided below.

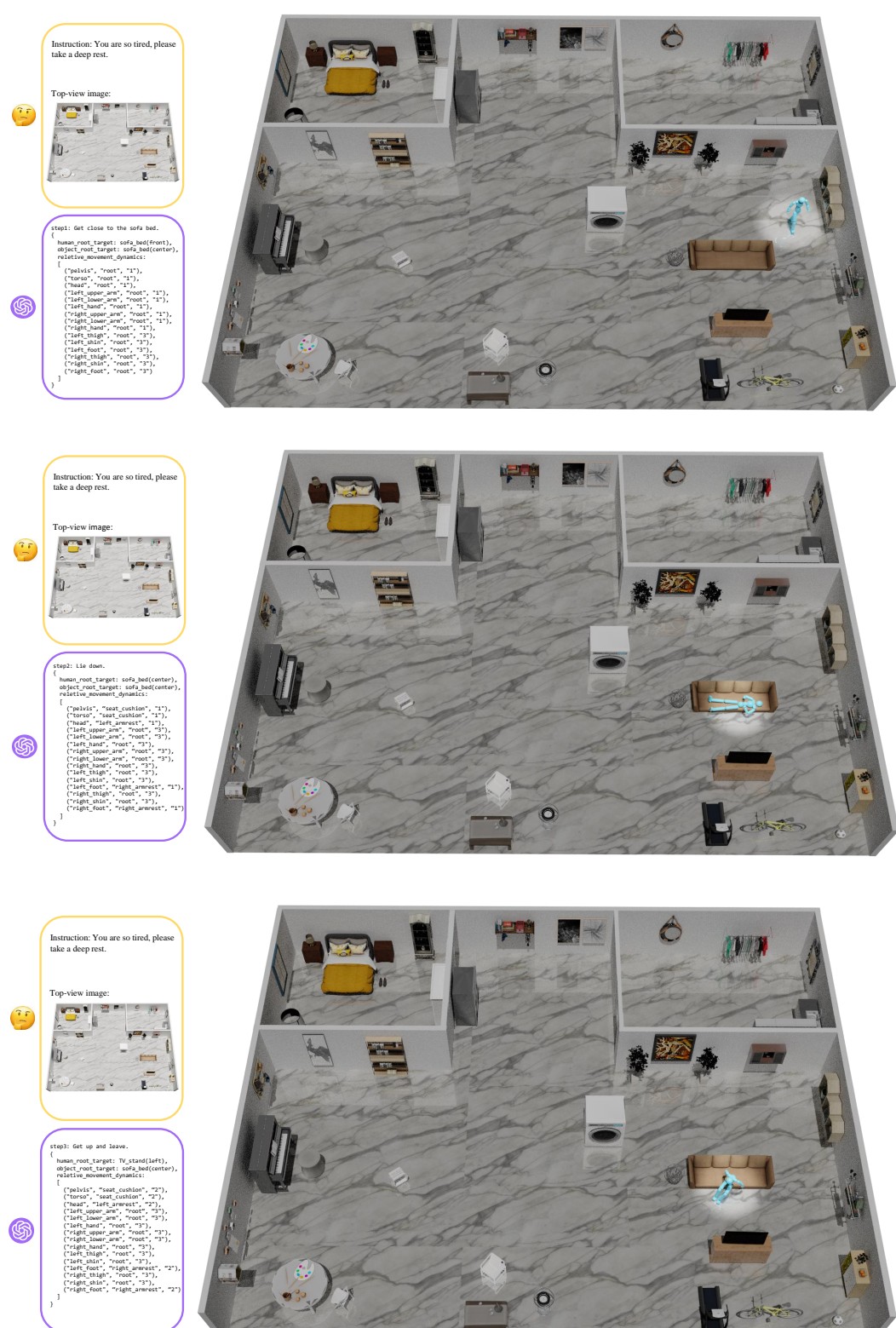

Figure 7: Visualization of long-term interaction with objects in an indoor home setting (part 1).

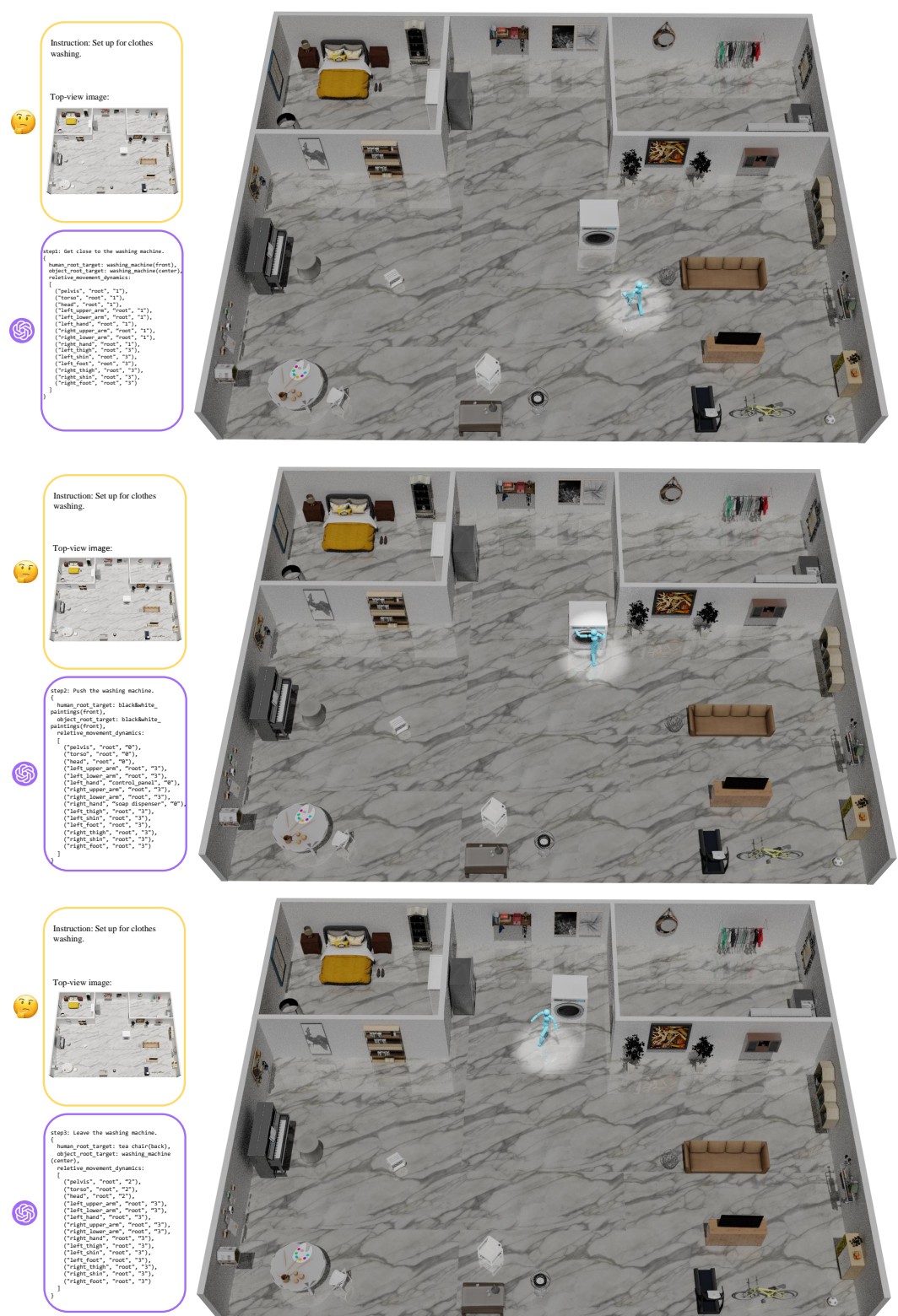

Figure 8: Visualization of long-term interaction with objects in an indoor home setting (part 2).

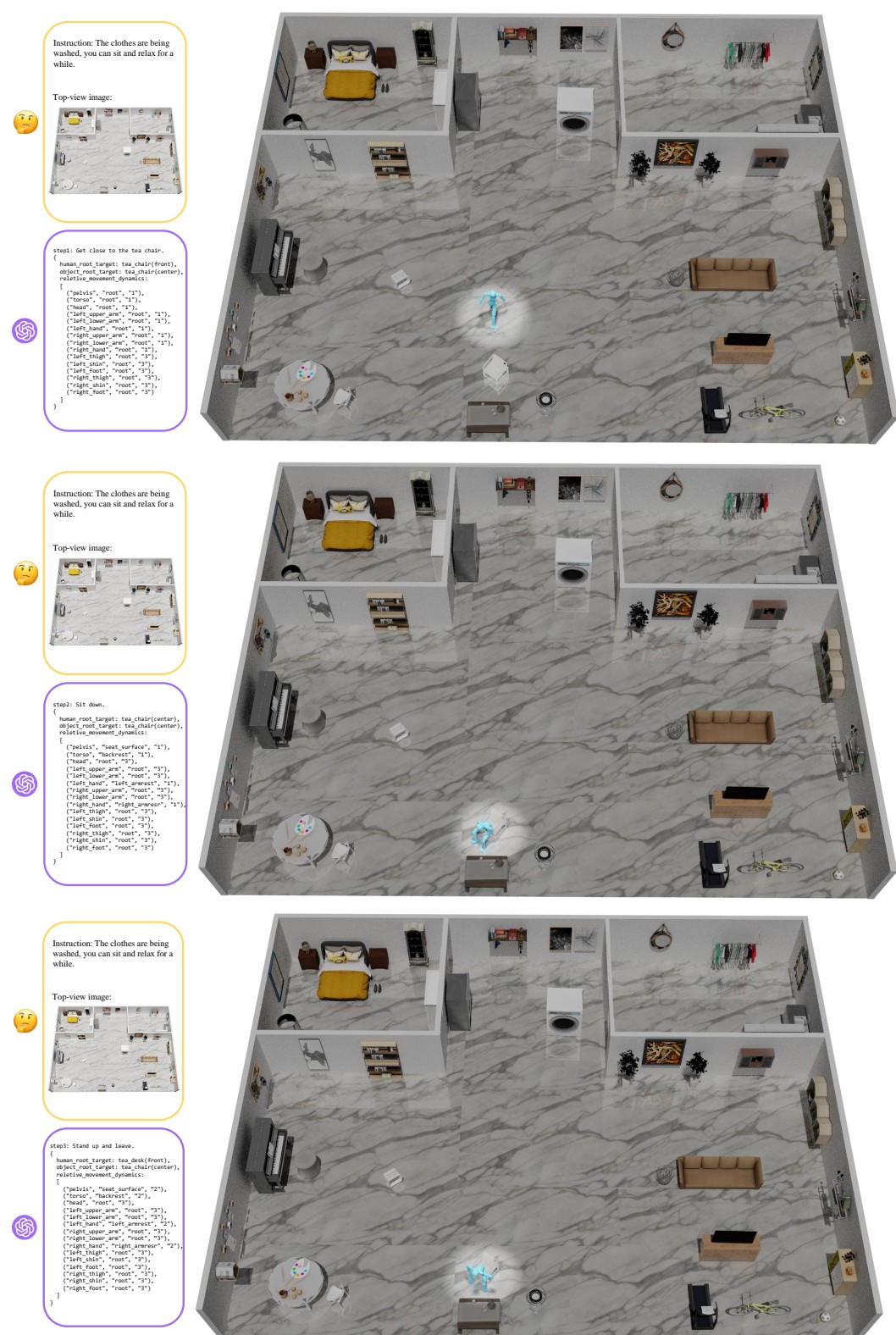

Figure 9: Visualization of long-term interaction with objects in an indoor home setting (part 3).

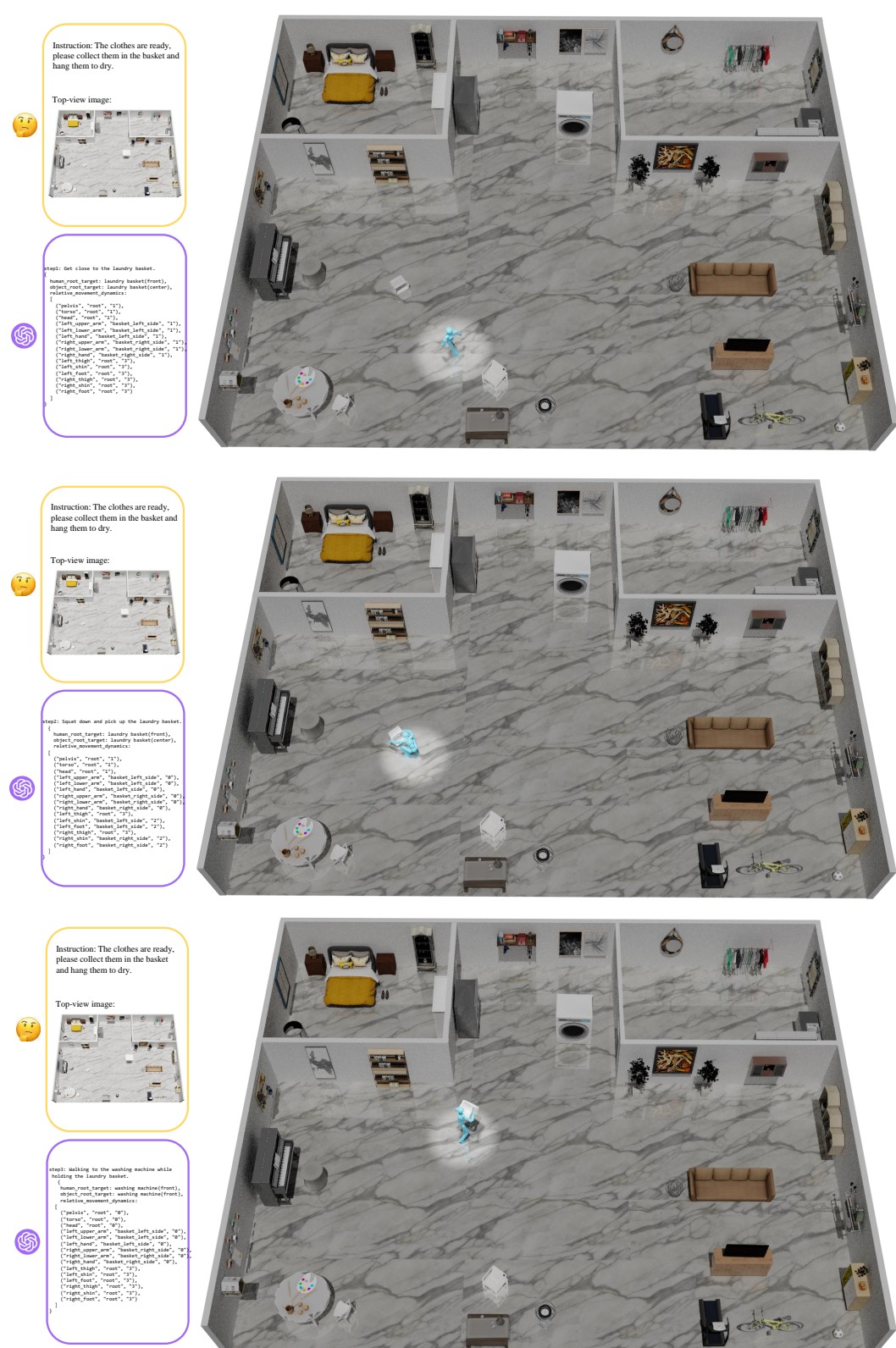

Figure 10: Visualization of long-term interaction with objects in an indoor home setting (part 4).

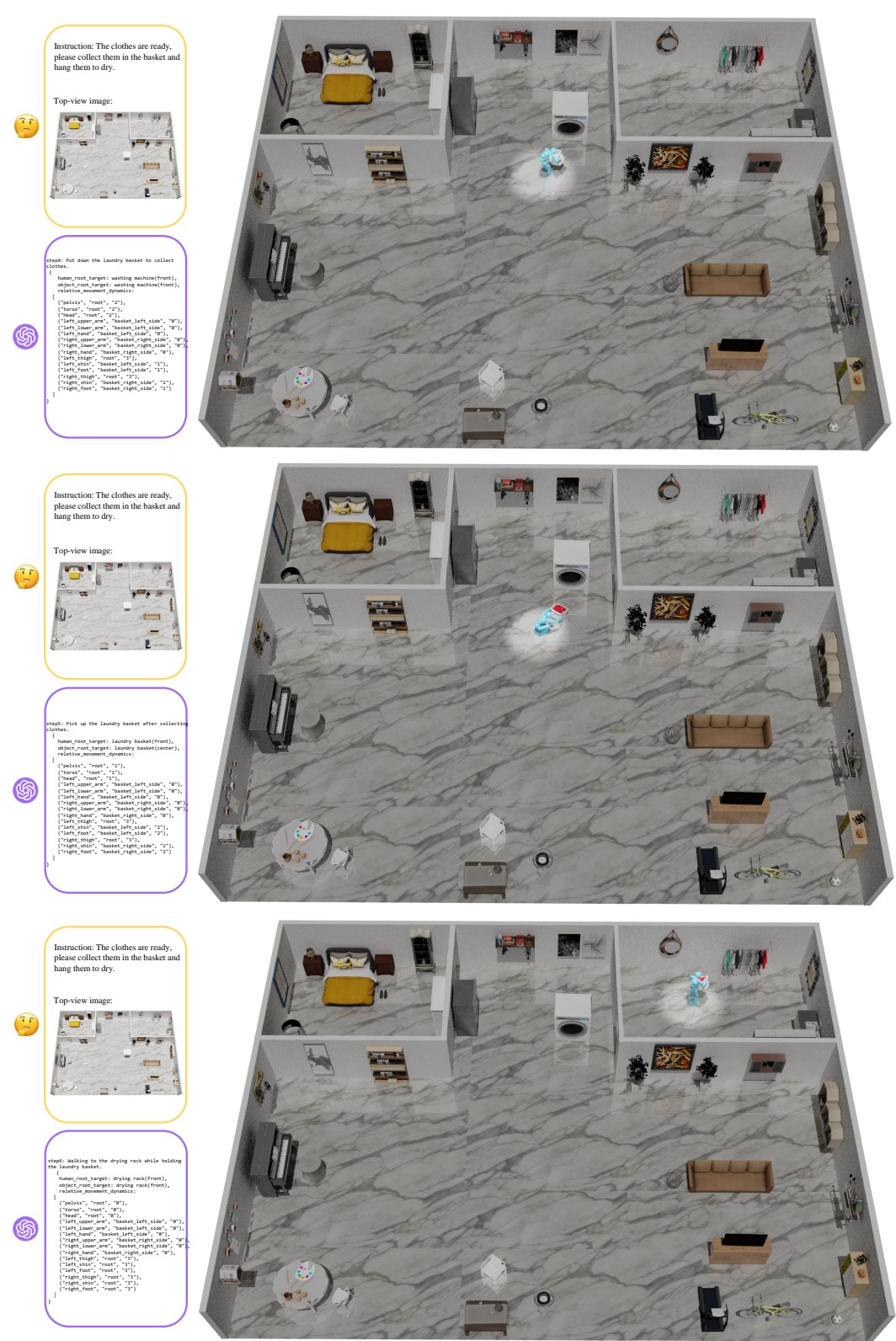

Figure 11: Visualization of long-term interaction with objects in an indoor home setting (part 5).

**Action.** We use a simulated humanoid character with 28 degrees of freedom (DoF), represented as a kinematic structure composed of 15 rigid bodies. The action space is defined as the target joint positions for 28 proportional–derivative (PD) controllers. At each timestep, the predicted actions specify these targets, which are then translated by the PD controllers into joint torques to drive the character's motion.

**Proprioception.** The proprioceptive state $\mathbf{s}_t$ is a 223-dimensional feature vector that encodes the character's physical state. It includes the positions, rotations, linear velocities, and angular velocities of all rigid bodies, expressed in the character's local coordinate frame. The only exception is the root height, which is represented in the world coordinate frame to preserve global elevation information.

**Task Reward.** Recall that the term $r_{\text{RMD}}$ encourages each edge $e_{ij} \in \mathcal{B}$, representing a *(human-part, object-part)* pair, to follow the relative motion dynamics specified by its associated edge weight $w_{ij}$, as planned by the VLM. Specifically, $r_{\text{RMD}}$ is computed by summing the individual alignment reward $r_{\text{rmd}}^{ij}(\cdot)$, each of which measures how well the pair $(p_{h_i}, p_{o_j})$ follows the motion pattern defined by $w_{ij}$, weighted by a coefficient $\lambda_{ij}$,

$$r_{\text{RMD}} = \sum_{(i,j)\in E} \lambda_{ij} \cdot r_{\text{rmd}}(\tilde{\mathbf{p}}_{ij}, \tilde{\mathbf{v}}_{ij}, w_{ij}). \tag{15}$$

Specifically, we formally define $r_{\text{rmd}}(\tilde{\mathbf{p}}_{ij}, \tilde{\mathbf{v}}_{ij}, w_{ij})$ as,

$$r_{\text{rmd}}(\tilde{\mathbf{p}}_{ij}, \tilde{\mathbf{v}}_{ij}, w_{ij}) = \begin{cases} \exp\left(-(\tilde{\mathbf{p}}_{ij} \cdot \tilde{\mathbf{v}}_{ij})^2\right), & w_{ij} = 0, \\[2ex] \frac{1}{2}\exp\left(-\|\tilde{\mathbf{p}}_{ij}\|_2^2\right) + \frac{1}{2}\exp\left(-\left(\tilde{\mathbf{v}}_{ij} \cdot \frac{\tilde{\mathbf{p}}_{ij}}{\|\tilde{\mathbf{p}}_{ij}\|_2} + v^\star\right)^2\right), & w_{ij} = 1, \\[2ex] \frac{1}{2}\left(1 - \exp\left(-\|\tilde{\mathbf{p}}_{ij}\|_2^2\right)\right) + \frac{1}{2}\exp\left(-\left(\tilde{\mathbf{v}}_{ij} \cdot \frac{\tilde{\mathbf{p}}_{ij}}{\|\tilde{\mathbf{p}}_{ij}\|_2} - v^\star\right)^2\right), & w_{ij} = 2, \\[2ex] 1, & w_{ij} = 3. \end{cases} \tag{16}$$

The behavior of each reward term varies depending on the value of $w_{ij}$:

- When $w_{ij} = 0$, the reward promotes either a vanishing relative velocity or an orthogonal orientation between the relative position and velocity, thereby minimizing directional alignment.

- For $w_{ij} = 1$, the goal is to maintain spatial proximity while aligning the relative velocity with the direction of the relative position. In particular, the velocity projection is encouraged to approach a predefined target $v^\star = 1\,\text{m/s}$.

- When $w_{ij} = 2$, the reward favors increasing separation between parts, while ensuring that the leaving speed along the direction of displacement remains close to $v^\star$.

- Finally, $w_{ij} = 3$ denotes an unstable or unpredictable relative motion trend. Here, the reward is set to a neutral constant value of 1 to avoid imposing misleading gradients.

**Training details in a single-task scenario.** Our policy network is divided into a critic network and an actor network, each starting with three multilayer perceptron (MLP) layers configured with [1024, 1024, 512] units. The discriminator network follows a similar architecture, also consisting of three MLP layers with [1024, 1024, 512] units. The hyperparameters used during the training process are detailed Tab. 4, Some baseline methods either did not implement certain HOI tasks (e.g., *open*, *push*) or failed to account for post-interaction transitions with static objects (e.g., *sit*, *lie*). To enable a fair and thorough comparison, we re-implemented these baselines with carefully hand-crafted goal states and reward functions for each task. This ensures that we can comprehensively evaluate the full interaction cycle: approaching the object, interacting with it, and departing afterward, across a range of individual HOI tasks. All methods were trained using a shared set of hyperparameters to maintain consistency.

**Training details in a long-horizon multi-task scenario.** Our policy consists of separate actor and critic networks, each implemented as a three-layer multilayer perceptron (MLP) with hidden dimensions of [1024, 1024, 512]. The discriminator network adopts the same architecture. Both our method and the baselines are trained using a two-stage procedure:

Table 4: Hyperparameters for RMD.

| Hyperparameter | Value | Hyperparameter | Value | Hyperparameter | Value |
|---|---|---|---|---|---|
| Num. envs | 4096 | Max episode length | 450 | Num. epochs | 25,000 |
| Discount factor $\gamma$ | 0.99 | GAE parameter $\lambda$ | 0.95 | Minibatch size | 16384 |
| Optimizer | Adam | Learning rate | 2e-5 | Gradient clip norm | 1.0 |
| PPO horizon length | 32 | PPO clip $\epsilon$ | 0.2 | PPO miniepochs | 6 |
| Actor loss weight | 1 | Critic loss weight | 5 | Discriminator loss weight | 5 |
| AMP batch size | 512 | AMP sequence length $t^*$ | 10 | AMP grad. penalty $w^{gp}$ | 5 |
| Task reward weight $\alpha_{\text{task}}$ | 0.5 | Style reward weight $\alpha_{\text{style}}$ | 0.5 | Bounds loss coefficient | 10 |

(i) *Single-task pre-training.* We divide the parallel simulation environments into distinct subsets according to task type and difficulty, allowing the agent to learn basic interaction skills in a focused manner.

(ii) *Long-horizon multi-task post-training.* For each environment instance, tasks are randomly sampled from our *InterPlay* dataset, promoting the agent's ability to handle long-horizon human–object interaction tasks in complex and diverse scenes.

To extend *InterPhys* to a multi-task setup, we concatenate all possible goal states and use a one-hot encoding to indicate the task identity. Additionally, we implement a script to translate the sequential plans in our dataset into the format required by other baselines (i.e., sequential goal states), thereby unifying the training pipeline. During execution, a rule-based finite-state machine is used to manage task transitions in both *InterPhys* and *TokenHSI*.

## D  VISUALIZATIONS IN SUPPLEMENTARY MATERIAL

To complement the qualitative results presented in the main paper, we provide a demonstration video that combines the key aspects of our method. This video offers detailed visualizations showcasing the effectiveness of our framework in various real-world scenarios and compares it with competing approaches. It includes an overview of the motivation, challenges, main pipeline, and both quantitative and qualitative results. Additionally, the video highlights long-horizon interactions in both single-task and multi-task scenarios, involving static, dynamic, and articulated objects within a realistic indoor setting. In the single-task scenario, we present a comparison of our method with UniHSI (Xiao et al., 2024) for the sitting task and InterPhys (Hassan et al., 2023) for the door-opening task. While UniHSI struggles with high-frequency jitter and unrealistic motion, our method ensures smooth, human-like transitions. Similarly, in the door-opening task, our approach uses coordinated hand movements for a more lifelike interaction, in contrast to InterPhys, which relies solely on body movement and produces unnatural results. The multi-task scenario illustrates how our method excels by leveraging human-like motion, a unified objective design across different tasks, and the capabilities of Vision-Language Models to generate contextually relevant action plans. Additionally, we provide detailed visualizations alongside the Planner's output, as shown in Fig. 7, Fig. 8, Fig. 9, Fig. 10 and Fig. 11.

## E  ABLATION STUDY ON THE VLM PLANNER

**Model and prompt variants.**  To assess the robustness of our approach, we performed ablation experiments by (i) replacing GPT-4V with other models such as LLaVA-1.6 and Qwen-VL-Max, and (ii) simplifying the prompt. As shown in Tab. 5, our framework remains effective across different VLMs and prompt designs, with only moderate performance differences. This is possible because our framework is modular, and the VLM planner functions as a plug-in component that can be replaced by any vision-language model with basic visual understanding and reasoning capabilities.

**Planner label accuracy on InterPlay.**  To further quantify planner reliability, we manually annotated ideal RMD labels for key human-object pairs across all tasks in each subset (static, dynamic, and hybrid) and compared them with the planner's outputs. The empirical rate of contradictory labels is $8.9\%$, $11.2\%$, and $14.7\%$ for static, dynamic, and hybrid tasks, respectively, with higher error

Table 5: Comparison of VLM choice and prompt designs.

| VLM | Completion Rate (%) ↑ | Sub-step Completion Ratio (%) ↑ | Sub-step Precision (cm) ↓ |
|---|---|---|---|
| LLaVA-1.6 | 43.3 | 67.8 | 13.1 |
| Qwen-VL-Max | 47.7 | 68.3 | 12.7 |
| GPT-4V (Simplified Prompt) | 45.3 | 66.1 | 12.9 |
| GPT-4V | 53.8 | 71.8 | 11.2 |

on hybrid tasks due to their complex multi-stage nature. Together with the ablations in Tab. 5, this analysis indicates that our framework is robust to moderate planner imperfections.

**Part decomposition on novel objects.** To further evaluate the planner's behavior on unseen assets, we additionally construct 100 new HOI tasks involving 50 objects that do not appear in the InterPlay dataset and manually inspect the VLM planner's part decompositions. Only 7 out of 100 cases exhibit clearly unreasonable part splits, suggesting that the planner generalizes reasonably well to unfamiliar but semantically common objects, likely due to its web-scale pretraining.

**Stage-transition threshold.** To evaluate the sensitivity of our method to the stage-transition threshold, we perform an ablation study on the InterPlay-Hybrid subset by sweeping the threshold from $0.80$ to $0.95$. As shown in Tab. 6, performance varies smoothly between $0.80$ and $0.90$ and peaks at $0.90$, while only the extreme setting $0.95$ leads to a clear drop in completion and precision. Based on these results, we keep $0.90$ as the default threshold, since it offers the best trade-off between completion, sub-step ratio, and precision while the range $[0.80, 0.90]$ is generally stable.

Table 6: Comparison of threshold choice.

| Threshold | Completion Rate (%) ↑ | Sub-step Completion Ratio (%) ↑ | Sub-step Precision (cm) ↓ |
|---|---|---|---|
| 0.80 | 47.9 | 65.6 | 13.5 |
| 0.85 | 51.0 | 67.2 | 12.9 |
| 0.90 | 53.8 | 71.8 | 11.2 |
| 0.95 | 32.8 | 47.2 | 16.7 |

# F  USER STUDY

To evaluate the visual quality and goal alignment of the generated motions from a human-centric perspective, we conducted a user study with 20 participants. Each participant watched 20 motion sequences, each paired with its corresponding task description, and rated them on two criteria: Motion Realism (assessing whether the interaction appears natural and physically plausible) and Task Consistency (measuring how well the motion aligns with the task objective). Each criterion was rated on a scale from 0 to 5, where higher scores indicate better performance. The results, summarized in Tab. 7, show that our method achieves superior performance in both realism and task alignment.

Table 7: Comparison of motion realism and task consistency.

| Method | Motion Realism ↑ | Task Consistency ↑ |
|---|---|---|
| InterPhys | $3.2 \pm 0.7$ | $3.5 \pm 0.5$ |
| TokenHSI | $3.4 \pm 0.4$ | $3.7 \pm 0.6$ |
| UniHSI | $3.1 \pm 0.5$ | $3.7 \pm 0.4$ |
| Ours | $\mathbf{4.0 \pm 0.4}$ | $\mathbf{4.1 \pm 0.3}$ |

## G STATEMENTS

**Ethics Statement.** This work complies with the ICLR Code of Ethics. The research involves no human subjects or sensitive personal data. All datasets are publicly available, and their use strictly follows the corresponding licenses and established ethical standards.

**Reproducibility Statement.** This work complies with the ICLR standards of reproducibility. We provide detailed descriptions of the experimental settings, model configurations, and evaluation protocols in the main text and appendix, ensuring that our results can be independently verified.

**LLM Usage Statement.** We disclose that large language models (LLMs) were used solely for grammar checking and minor text polishing. All aspects of the research design, experimentation, analysis, and results remain entirely the responsibility of the authors.

