# OpenReview forum: "Human-Object Interaction via Automatically Designed VLM-Guided Motion Policy"
_ICLR.cc/2026/Conference — ICLR 2026 Poster_

### Official Review · Reviewer_iGBJ · 2025-10-31

**Soundness:** 3
**Presentation:** 3
**Contribution:** 3
**Rating:** 6
**Confidence:** 4

**Summary:**

For HOI synthesis, this paper’s main contributions are twofold: (1) using a VLM as a high-level planner and (2) proposing RMD as a mid-level interface between planning and control. The method requires neither task-specific motion-capture demonstrations nor manual reward engineering, and supports long-horizon, multi-stage interactions with static objects, dynamic objects, and articulated objects. The authors also introduce InterPlay, a dataset containing thousands of long-horizon interaction plans for systematic evaluation. Experiments show consistent gains over strong baselines (e.g., AMP, InterPhys, UniHSI, TokenHSI) in completion rate and precision across single-task and multi-task settings. Ablations further demonstrate that VLM visual input and the fine-grained RMD representation are key to the performance.

**Strengths:**

* Comprehensive ablations and analysis. The paper isolates component contributions via ablations and alternative VLMs. Removing visual input or coarsening RMD degrades performance, underscoring the importance of VLM spatial grounding and fine-grained RMD guidance. A user study in the appendix aligns with quantitative metrics, indicating more natural and plausible motions.

* Thorough empirical validation. InterPlay spans diverse scenes and task types. Metrics (e.g., Completion Rate, Precision) are appropriate, and comparisons against representative baselines cover static, dynamic, and multi-task scenarios. Results show sizable improvements (often tens of percentage points in completion) and higher motion fidelity, supporting generalization and robustness.

**Weaknesses:**

* Planner reliability. While GPT-4V outputs are structured, the planner remains a black box. Erroneous sub-plans could hinder training or misguide rewards (e.g., incorrect reasoning about dynamic object trends). The paper does not quantify or correct such errors. Moreover, cross-phase physical causality is only implicitly encoded via RMD; the VLM may be pattern-matching rather than reasoning about outcomes, which could break in highly dynamic tasks (e.g., throwing/catching).

* RMD’s bipartite modeling misses self-contact and intra-object contact. By construction, RMD only encodes relations between human parts and object parts. Many everyday interactions require human–human self-contact and articulated object self-contact (e.g., folding a clamshell laptop). Such intra-agent and intra-object constraints are not representable with a strictly bipartite graph, which can lead to inaccurate guidance or reward shaping for these tasks.

* Task scope. Extremely dynamic, high-precision, or fast-response scenarios are not covered; it is unclear whether RMD + GPT-4V planning scales to these regimes.

* Writing. The related-work narrative around “using a planner” vs. “the representation that the planner outputs” is not sufficiently disentangled; these are distinct axes and should be contrasted more clearly.

**Questions:**

* Physical causal reasoning. The method relies on VLM-imagined motion trends. In highly dynamic settings (e.g., projectile interactions or multiple moving objects), can GPT-4V reason causally? Can RMD be extended to handle exogenous object motion (not initiated by the agent)? If objects move autonomously or unexpected events occur (e.g., a rolling ball), can the framework revise the plan online?

* Planner stability. Did you observe VLM hallucinations (e.g., incorrect part segmentation, anomalous RMD weights)? What filtering or correction mechanisms were used? How sensitive are plans to prompt phrasing? Please provide observations and mitigation strategies for planner stability.

* Open resources. Please confirm the release timeline for the InterPlay dataset and the codebase (including prompt templates and exemplar plans).

---

> ### Author Response · Authors · 2025-11-23
> **PART 1/3**
>
> Thank you for your valuable comments and feedback.
>
> >W1: Planner reliability. While GPT-4V outputs are structured, the planner remains a black box. Erroneous sub-plans could hinder training or misguide rewards (e.g., incorrect reasoning about dynamic object trends). The paper does not quantify or correct such errors. Moreover, cross-phase physical causality is only implicitly encoded via RMD; the VLM may be pattern-matching rather than reasoning about outcomes, which could break in highly dynamic tasks (e.g., throwing/catching).
>
>
> To quantify planner reliability, we evaluated all tasks in each subset (static / dynamic / hybrid), manually annotated ideal RMD labels for key human–object pairs, and compared them with the planner’s outputs. The rate of contradictory labels is 8.9% / 11.2% / 14.7% for static / dynamic / hybrid, with higher error on hybrid tasks due to their multi-stage interactions. Moreover, our existing empirical results as reported in Tab. 5, already provide indirect evidence of robustness to planner imperfections.
>
> We argue that our method **does perform physical reasoning**, rather than simple pattern matching, for several key reasons. First, the VLM's inherent world knowledge allows it to decompose high-level instructions into semantically meaningful sub-tasks and generate corresponding motion graphs, as illustrated in Appendix A, Fig. 6. This structured planning process demonstrates a form of motion reasoning.
>
> Regarding highly dynamic tasks like throwing/catching, we agree they present a challenging frontier. These tasks are highly sensitive to micro-phases and require finely-tuned, specific reward functions. While our current method focuses on a broader set of manipulation tasks, extending it to such dynamic scenarios is a compelling and important direction for our future work.
>
> ---
>
> >W2: RMD’s bipartite modeling misses self-contact and intra-object contact. By construction, RMD only encodes relations between human parts and object parts. Many everyday interactions require human–human self-contact and articulated object self-contact (e.g., folding a clamshell laptop). Such intra-agent and intra-object constraints are not representable with a strictly bipartite graph, which can lead to inaccurate guidance or reward shaping for these tasks.
>
> Thank you for the insightful question. The bipartite design of RMD represents a trade-off between performance and complexity and is further constrained by the limited output token speed of our VLM-based planner. Modeling all possible contacts would lead to an exponential increase in complexity. We fully agree that modeling self-contact and intra-object articulation is essential for extending the approach to a wider range of tasks. Importantly, our RMD graph is designed to be easily extensible, and incorporating explicit human-human and object-object interactions represents a promising direction for future work.
>
> ---
>
> >W3: Task scope. Extremely dynamic, high-precision, or fast-response scenarios are not covered; it is unclear whether RMD + GPT-4V planning scales to these regimes.
>
> Our work focuses on everyday human-object interactions with moderate dynamic or precision, such as transporting an armchair or operating a door. As the reviewer notes, extremely dynamic or high-precision tasks (e.g., running downstairs while carrying a full bowl of soup) present a fundamentally different set of challenges. These are constrained not only by planning but critically by hardware limits and the precision and latency of low-level control. The current robotics and RL community still lacks general, robust solutions for such regimes. In these challenging scenarios, we believe RMD could still provide a high-level compositional structure for the interaction. However, ultimate success would depend on a specialized, high-performance low-level controller (e.g., high-frequency MPC or hierarchical RL). Integrating RMD with such controllers for extreme tasks is beyond the scope of this paper but represents a compelling direction for future work.

---

> ### Author Response · Authors · 2025-11-23
> **PART 2/3**
>
> >W4: Writing. The related-work narrative around “using a planner” vs. “the representation that the planner outputs” is not sufficiently disentangled; these are distinct axes and should be contrasted more clearly.
>
> We thank the reviewer for this valuable feedback. In our revision, we will explicitly separate the discussion of planners from their output representations to better frame the related work part.
>
> In physics-based HOI, some recent works start to use external planners to decompose complex tasks into simpler sub-tasks. InterScene [2] uses a rule-based action scheduler to turn a high-level interaction into a sequence of reusable low-level policies, and HOI-FHLI [5] employs an LLM planner that outputs textual sub-task descriptions and target object poses. In both cases, the planner mainly performs task decomposition and provides limited additional guidance to the controller. More recent work, such as UniHSI [1] makes the planner’s output representation itself a key contribution: HOI tasks are translated into a structured, semantically rich Chain-of-Contacts used for goal and reward design. However, UniHSI is restricted to static HOI, its planner does not leverage VLM-based motion imagination, and its representation does not capture motion dynamics. In the broader RL literature, Eureka [3] and GROVE [4] also employ LLMs as planners, but the guidance they provide to the policy is solely through automatically designed reward functions, rather than through a structured mid-level HOI representation like our RMD. Our method lies at the intersection of these axes: we use a VLM planner and introduce a planner-generated RMD representation that encodes part-level relative movement, providing fine-grained guidance for physics-based HOI control.
>
> ---
>
> >Q1: Physical causal reasoning. The method relies on VLM-imagined motion trends. In highly dynamic settings (e.g., projectile interactions or multiple moving objects), can GPT-4V reason causally? Can RMD be extended to handle exogenous object motion (not initiated by the agent)? If objects move autonomously or unexpected events occur (e.g., a rolling ball), can the framework revise the plan online?
>
>
> **On VLM Causal Reasoning in Highly Dynamic Settings**: We intentionally scoped our study to moderately dynamic human-object interactions (e.g., carrying, pushing). As the reviewer rightly points out, highly dynamic scenarios (e.g., involving projectiles) would require reasoning over many short sub-phases and intricate physical causality, which we agree is likely beyond the reliable capabilities of current VLMs. Tackling such extreme settings is an important but challenging direction for future research.
>
> **On Handling Exogenous Object Motion**: Yes, our RMD formulation can naturally handle exogenous object motion. The key is that RMD is defined purely over the observed relative positions and velocities between human and object parts, making it agnostic to whether the motion is agent-initiated or externally caused. To empirically verify this, we conducted two new experiments without any modifications to the RMD graph: (1) An auto-closing door with a hinge that applies a restoring torque. (2) A drifting box that moves at a constant speed in a random direction until grasped. The learned policies achieved success rates of 90.4% and 84.8% respectively, demonstrating the method's effectiveness under exogenous dynamics.
>
> **On Online Replanning**: In our current implementation, the VLM planner produces an offline RMD plan at the start of an episode. While the low-level policy is closed-loop and reacts to the state, the high-level RMD sequence is fixed. Importantly, our architecture can naturally support online replanning at a low additional cost. By re-querying the VLM with an updated observation, a new RMD plan can be generated and fed into the same goal-conditioned policy. Implementing and evaluating such explicit online replanning for large unforeseen disturbances is a compelling direction for future work.

---

> ### Author Response · Authors · 2025-11-23
> **PART 3/3**
>
> >Q2: Planner stability. Did you observe VLM hallucinations (e.g., incorrect part segmentation, anomalous RMD weights)? What filtering or correction mechanisms were used? How sensitive are plans to prompt phrasing? Please provide observations and mitigation strategies for planner stability.
>
> We do observe occasional VLM hallucinations in practice, such as slightly noisy edge weights or minor inconsistencies in part assignments. However, our quantitative evaluation (detailed in Response W1) demonstrates that these occurrences are limited and the planner remains overall **stable**.
>
> To mitigate these issues, we have implemented several explicit safeguards: (1)As illustrated in Fig. 5, our "Plan Rules" explicitly instruct the planner to **fall back** to coarse part segmentation and mark edges as unstable when confidence is low, preventing the generation of confident but incorrect plans. (2) During the construction of InterPlay, we performed light manual **filtering** to remove plans with obvious hallucinations, ensuring severe errors rarely propagate to policy training.
>
> We further evaluate the sensitivity of the planner to both VLM choice and prompt phrasing. As reported in Tab. 5,
> replacing GPT-4V with other VLMs, or simplifying the planner prompt, causes only moderate changes in completion rate and precision, indicating that the framework is not overly sensitive to specific model choices or prompt wording.
>
> ---
>
> >Q3 Open resources. Please confirm the release timeline for the InterPlay dataset and the codebase (including prompt templates and exemplar plans).
>
> We will release the full codebase and the InterPlay dataset upon paper acceptance. The release will include all necessary components: the processed 3D assets, scene layouts, VLM-generated RMD plans, prompt templates, exemplar plans, and trained model checkpoints. All materials will be hosted permanently on our project website and GitHub repository to ensure accessibility and support reproducibility.
>
> ---
>
> **References**
> >[1] Unified Human-Scene Interaction via Prompted Chain-of-Contacts.
>
> >[2] Synthesizing Physically Plausible Human Motions in 3D Scenes.
>
> >[3] Eureka: Human-Level Reward Design via Coding Large Language Models.
>
> >[4] GROVE: A Generalized Reward for Learning Open-Vocabulary Physical Skill.
>
> >[5] Human-Object Interaction from Human-Level Instructions.

---

### Official Review · Reviewer_7XuJ · 2025-10-31

**Soundness:** 3
**Presentation:** 3
**Contribution:** 3
**Rating:** 8
**Confidence:** 4

**Summary:**

This paper addresses the limitations of existing Human-Object Interaction (HOI) synthesis methods—such as reliance on expensive motion capture data, manual reward engineering, and poor support for long-horizon/dynamic interactions—by proposing a unified physics-based framework. The core innovation is VLM-Guided Relative Movement Dynamics (RMD), a fine-grained spatio-temporal bipartite graph representation that encodes part-level relationships between humans and objects (e.g., stationary contact, approaching/separating motion). RMD enables Vision-Language Models (VLMs, e.g., GPT-4V) to automatically generate goal states and reward functions for reinforcement learning (RL), eliminating manual tuning. The framework supports interactions with static, dynamic, and articulated objects, and the authors introduce InterPlay, a novel dataset of thousands of long-horizon HOI plans. Experiments demonstrate that the method outperforms baselines (e.g., InterPhys, TokenHSI, UniHSI) in both single-task and long-horizon multi-task scenarios, producing more natural, task-aligned motions as validated by quantitative metrics and a user study.

**Strengths:**

1. Captures fine-grained spatio-temporal dynamics between human and object parts, addressing the limitations of coarse contact-based (e.g., chain-of-contacts) or kinematic-only representations. This enables modeling of continuous interaction dynamics (e.g., carrying an object) rather than discrete events.

2.  Leverages VLMs to translate high-level instructions and scene context into structured RMD plans, which are directly mapped to goal states and composite rewards. This eliminates labor-intensive manual reward engineering, a major bottleneck for scalable HOI.

3. Uniquely supports static, dynamic, and articulated objects, as well as long-horizon multi-task sequences (e.g., pick up → carry → place → sit). Most prior methods are limited to single-task or static-only interactions.

4.  Fills a critical gap in existing HOI datasets by providing long-horizon, context-rich interaction plans, enabling systematic evaluation of multi-task HOI synthesis.

3. Outperforms baselines across key metrics (completion rate, sub-step precision) in both single and multi-task settings. The user study further confirms superior motion naturalness and task alignment.

**Weaknesses:**

1. The framework only supports single-agent interactions, ignoring multi-agent collaboration (e.g., two people moving a sofa) or social dynamics—key for real-world applications like assistive robotics or collaborative environments.

2. The VLM planner may struggle with extremely complex long-horizon tasks requiring deep hierarchical planning (e.g., multi-step cooking with ingredient prep, cooking, and serving), as noted in the paper’s future work.

3. Object part decomposition depends on the VLM’s implicit judgment, with no explicit evaluation of decomposition accuracy or robustness to novel/unfamiliar objects (e.g., specialized tools).

**Questions:**

1. The edge weight categories (0=stationary, 1=approaching, 2=separating, 3=unstable) are heuristic—what empirical or theoretical basis supports this granularity? Could a learned weight space or more granular categories (e.g., varying speed of approach) improve interaction naturalness?

2. How does the VLM handle part decomposition for novel objects with ambiguous functional parts (e.g., a multi-purpose tool)? Is there a fallback mechanism, and how is decomposition quality evaluated?

3. For tasks more complex than those in InterPlay (e.g., "tidy the living room, wash dishes, and prepare coffee"), how does the framework scale? Would integrating chain-of-thought prompting (mentioned in future work) quantitatively improve plan coherence and task completion?

---

> ### Author Response · Authors · 2025-11-23
> **PART 1/2**
>
> We are grateful for your detailed feedback, which helped us improve the clarity and quality of the paper.
>
> >W1: The framework only supports single-agent interactions, ignoring multi-agent collaboration (e.g., two people moving a sofa) or social dynamics—key for real-world applications like assistive robotics or collaborative environments.
>
>
> We agree that multi-agent collaboration and social dynamics are crucial for real-world applications. We have already acknowledged this in our Limitations and Future Work section (Lines 1374–1378, 1383–1385). While the current work deliberately focuses on the single-agent setting, both the VLM-guided planner and the RMD representation are agent-agnostic and can be naturally **extended** to multiple humanoids—for example, by instantiating multiple human node sets in the RMD graph. Exploring such multi-agent extensions and socially aware coordination is an important direction that we plan to pursue in future work.
>
> ---
>
> >W2: The VLM planner may struggle with extremely complex long-horizon tasks requiring deep hierarchical planning (e.g., multi-step cooking with ingredient prep, cooking, and serving), as noted in the paper’s future work.
>
> We agree that extremely complex, deeply hierarchical tasks (e.g., full cooking workflows) remain challenging for current VLM planners. Our work focuses on **long-horizon** HOI sequences involving **multiple objects** and sub-goals, where the proposed VLM-guided planner already demonstrates **strong performance**. In the near future, techniques such as RAG and ToT, which enhance knowledge retrieval and reasoning ability, will be instrumental. Extending the planner with more explicit hierarchical decomposition for recipe-level workflows will be a natural and promising direction for future work.
>
>
> ---
>
> >W3: Object part decomposition depends on the VLM’s implicit judgment, with no explicit evaluation of decomposition accuracy or robustness to novel/unfamiliar objects (e.g., specialized tools).
>
> We thank the reviewer for the question. To directly evaluate decomposition quality on novel objects, we construct 100 new tasks with 50 objects that do not appear in InterPlay, and manually inspect the VLM planner’s part decompositions. Only **7/100** cases show clearly unreasonable part splits, suggesting that the planner handles unfamiliar but semantically common objects reasonably well, likely due to its broad web-scale pretraining.
>
> In addition, our prompt design explicitly mitigates decomposition errors. As shown in Fig. 5, the “Plan Rules” instruct the planner to fall back to the object root when it is unsure about fine-grained parts, leading to conservative but physically meaningful interactions rather than unrealistic ones. We further probe robustness to part granularity via the “multi-one” ablation in Tab. 3, where all parts are collapsed to a single root node: performance degrades moderately but does not collapse, indicating that our framework is **tolerant** to imperfect or coarse part decompositions.

---

> ### Author Response · Authors · 2025-11-23
> **PART 2/2**
>
> >Q1: The edge weight categories (0=stationary, 1=approaching, 2=separating, 3=unstable) are heuristic—what empirical or theoretical basis supports this granularity? Could a learned weight space or more granular categories (e.g., varying speed of approach) improve interaction naturalness?
>
>
> Thank you for this insightful question. Our design is grounded in both empirical validation and a balance of competing requirements.
>
> First, we empirically validated the importance of our graph structure through an ablation study (Table 3), which shows that our chosen connectivity is necessary for strong performance, as simplified variants lead to a noticeable drop. Regarding the 4-category edge weights, this is a principled trade-off between **expressiveness, robustness, and planner compatibility**.
>
> **Expressiveness**: The three core semantic labels (stationary, approaching, separating) are sufficient to capture the dominant, distinguishable modes of interaction (e.g., "stationary" for support, "approaching" for manipulation). The fourth "unstable" category acts as a crucial safety buffer for ambiguous or noisy kinematic signals.
>
> **Robustness**: Finer-grained schemes (e.g., based on speed) are brittle, as low-level kinematic data from physics simulators is often too noisy to support such distinctions reliably. Our discrete set provides a robust abstraction.
>
> **Planner Compatibility**: A learned continuous space would sever the interpretable, symbolic link to the VLM, making it difficult for the planner to generate and reason about goals. Our discrete labels are compact enough for the VLM to predict reliably while being rich enough to drive diverse and physically plausible motions.
>
> Thus, our design is the optimal compromise for a hierarchical VLM-to-control framework, though exploring richer yet planner-compatible representations is a compelling direction for future work.
>
> ---
>
> >Q2: How does the VLM handle part decomposition for novel objects with ambiguous functional parts (e.g., a multi-purpose tool)? Is there a fallback mechanism, and how is decomposition quality evaluated?
>
> Thank you for this excellent point. Our approach handles this challenge through a combination of the VLM's inherent semantic understanding and a designed fallback mechanism.
>
> **Handling Ambiguity via Semantic Priors**: The VLM's web-scale pretraining provides a strong prior for common object structures. It can dynamically activate different parts of an object based on the task phase. For instance, as shown in the video (4:50–5:20), for an armchair: the "armrests" are primarily activated for the "move" task, while "sitting" activates the "seat" and "backrest." This demonstrates a contextual understanding similar to handling a multi-purpose tool.
>
> **Lightweight Fallback Mechanism**: We intentionally avoid a complex, learned "judge" module to maintain system simplicity. Instead, we implemented a prompt-level fallback (Fig. 5): if the VLM is uncertain about fine-grained parts, our instructions allow it to select the object's root node. This results in a conservative but physically plausible interaction with the whole object, ensuring robustness.
>
> **Evaluation of Decomposition Quality**: We directly evaluated this by testing the VLM planner on 100 novel tasks involving 50 objects not in InterPlay. Through manual inspection, we found only 7/100 cases had clearly unreasonable part decompositions. This high success rate, combined with the empirical robustness to planning errors noted in our response to W3, confirms that the decomposition is generally reliable for semantically common objects.
>
> ---
>
> >Q3: For tasks more complex than those in InterPlay (e.g., "tidy the living room, wash dishes, and prepare coffee"), how does the framework scale? Would integrating chain-of-thought prompting (mentioned in future work) quantitatively improve plan coherence and task completion?
>
> Thanks for the question. InterPlay already targets long-horizon, multi-step HOI with multiple objects. Scaling to even more complex tasks that span multiple rooms/scenes and involve dozens of sub-goals mainly requires stronger high-level task decomposition rather than changes to our mid-level RMD or low-level policies. Our framework is compatible with hierarchical planners and chain-of-thought (CoT) prompting: for a task such as “tidy the living room”, a CoT-augmented VLM can first decompose the instruction into a sequence of mid-level sub-goals (e.g., “place vase on shelf A”, “move armchair to table B”, …), and then convert each sub-goal into an RMD plan while explicitly checking spatial relations and feasibility. This step-by-step decomposition and validation is expected to improve the accuracy and stability of very long-horizon plans, and a systematic quantitative study of CoT-based planning within our framework is an interesting direction for future work.

---

### Official Review · Reviewer_xkXT · 2025-11-01

**Soundness:** 3
**Presentation:** 3
**Contribution:** 3
**Rating:** 8
**Confidence:** 4

**Summary:**

The paper proposes a unified physics-based HOI framework that uses a VLM (GPT-4V) to automatically construct both goal states and reward functions from a structured representation called Relative Movement Dynamics (RMD). RMD is a part-to-part bipartite graph over human and object components with discrete relative-motion labels (e.g., approach, separate, stationary ). The VLM planner outputs a sequence of RMD steps plus spatial anchors; the RL policy (PPO) then executes them with a composite task reward (human/object destination + RMD consistency) and a style prior. A new InterPlay dataset supports long-horizon static, dynamic, and articulated interactions. Experiments show large gains in completion, sub-step ratio, and precision, with ablations that isolate the value of VLMs, multi-part modeling, and the RMD features.

**Strengths:**

- The RMD abstraction gives the VLM a concrete “language” to plan in, and gives RL a direct mapping from that plan to goals and rewards. This avoids per-task reward tuning and makes long-horizon transitions feel principled instead of different parts being glued together. This is a meaningful step toward building more general-purpose simulated agents.
- I appreciate the completion definition (do the interaction and return to a neutral position). It addresses what usually breaks in HOI: recovery and chaining. The method’s advantage in hybrid / multi-task settings (Table 2) is meaningful.
- The Interplay dataset, including both static and dynamic objects and requiring multi-object sequences, is a welcome addition to the community.

**Weaknesses:**

- In the demo video opening door scene, it is actually very hard to see the agent using its hand/stump.
- The system leans heavily on GPT-4V with prompts.  What is the failure case for GPT-4V that may impact training and learning agents?
- There are two compute costs here: (i) planner inference (VLM) and (ii) training/execution. It would help to see end-to-end wall-clock and per-episode runtime (planning + control), and how that compares to baselines.
- The naturalness of the human motion ultimately leans on known motion-style priors (AMP-like discriminator), which means out-of-distribution motion may not be feasible.

**Questions:**

- What is the empirical error rate of the VLM planner on InterPlay, and how does task performance degrade?
- The stage transition uses a fixed and hand-picked 0.9 threshold. How sensitive are completion and precision to this value, and did you try progress-based or confidence-weighted switching that might be less brittle in cluttered or dynamic cases?

**Details Of Ethics Concerns:**

- What is the empirical error rate of the VLM planner on InterPlay and how does task performance degrade?
- The stage transition uses a fixed and hand-picked 0.9 threshold. How sensitive are completion and precision to this value, and did you try progress-based or confidence-weighted switching that might be less brittle in cluttered or dynamic cases?

---

> ### Author Response · Authors · 2025-11-23
> **PART 1/2**
>
> Thank you for taking the time to review our work and provide insightful suggestions.
>
> >W1: In the demo video opening door scene, it is actually very hard to see the agent using its hand/stump.
>
> Thank you for pointing this out. We have updated the supplementary video with a close-up visualization that clearly highlights the hand-door interaction for the "open-the-door" task. The new footage demonstrates that our policy successfully employs both hands to push the door open in a natural, human-like manner.
>
> ---
>
> >W2: The system leans heavily on GPT-4V with prompts. What is the failure case for GPT-4V that may impact training and learning agents?
>
> The motion policies in InterPlay are trained on a manually cleaned set of plans, where invalid or physically inconsistent outputs are filtered out. This ensures the agent learns from high-quality data. More importantly, as shown in Table 5, our system demonstrates significant robustness: when we replace GPT-4V (53.8% success) with weaker VLMs (43.3%, 47.7%) or use simplified prompts (45.3%), performance degrades only moderately rather than collapsing. This indicates that the learned policy is not brittle and can compensate for reasonable levels of planner noise.
>
> A typical failure mode involves errors in fine-grained spatial reasoning. For instance, when tasked with "returning the armchair to its position near the dining table," the VLM planner might designate a target location that is too narrow. The constrained space makes it physically impossible for the agent to maneuver and place the armchair at the specified location while carrying it.
>
> ---
>
> >W3: There are two compute costs here: (i) planner inference (VLM) and (ii) training/execution. It would help to see end-to-end wall-clock and per-episode runtime (planning + control), and how that compares to baselines.
>
> Thanks. Our system's runtime is best analyzed in two distinct phases, as the planner and controller are temporally decoupled.
>
> **Planner cost.** The RMD planner runs offline and is executed only once to generate the full sequence of sub-goals in the form of RMD. A single VLM call takes approximately 8s to produce the complete plan. The resulting RMD plans are cached and reused throughout training and evaluation, so planner inference is never invoked in the online control loop and imposes no runtime overhead during policy execution. The planner cost is comparable to baseline like UniHSI.
>
> **Control training and per-episode runtime.** The downstream controller is a lightweight MLP policy running at 30 Hz inside the NVIDIA IsaacGym simulator. For a single HOI task such as carry, one rollout consists of about 300 timesteps (≈10 s simulated duration), and we train each task for roughly 25K iterations. The training and inference latency of this control policy is essentially the same as for other RL baselines using the same simulator and architecture.
>
> ---
>
> >W4: The naturalness of the human motion ultimately leans on known motion-style priors (AMP-like discriminator), which means out-of-distribution motion may not be feasible.
>
> We thank the reviewer for this point and wish to clarify that our method's strength lies precisely in its ability to generate feasible out-of-distribution motions, which is a direct result of our core design. Unlike tracking-based approaches that mimic existing clips, our policy is trained under a **goal-conditioned RL** framework. It is explicitly optimized to achieve the fine-grained spatial goals (RMD) proposed by the VLM Planner. During training, the environment is randomized with varying object positions, scales, and poses, forcing the policy to learn a robust, generalizable strategy for goal achievement rather than mimicking a specific trajectory. As shown in Eq. 14, the AMP-like discriminator acts as a secondary motion-style prior. Its sole function is to ensure that the diverse motions discovered by the RL policy are kinematically plausible and human-like, consistent with the mocap data. It does not dictate what action to take, but rather how that action should be executed. Therefore, the **generalization capability for OOD tasks** originates from our **VLM and RMD design**, while the discriminator simply guarantees the naturalness of the resulting motion.

---

> ### Author Response · Authors · 2025-11-23
> **PART 2/2**
>
> >Q1: What is the empirical error rate of the VLM planner on InterPlay, and how does task performance degrade?
>
> To address the reliability of the VLM planner, we conducted a dedicated evaluation by manually annotating ground-truth RMD labels for all tasks in the InterPlay dataset. The analysis reveals an empirical error rate (quantified as contradictory labels) of 8.9% for static, 11.2% for dynamic, and 14.7% for hybrid tasks. As anticipated, error rates are highest for hybrid tasks due to their complex, multi-stage nature.
>
> Critically, our full-system performance remains high (Table 2) due to the inherent robustness of our policy training. The low-level RL policy is trained to handle goal variations, where a few incorrect RMD edges from the planner act as minor noise within a comprehensive reward function that combines accurate root targets with multiple other terms and adaptive weights. This resilience is further validated empirically in Table 5 (Ablation: "Weakened Planner"), where the deliberate introduction of planner noise results only in a smooth, proportional performance drop, confirming the system's robustness to planner imperfections.
>
> ---
>
> >Q2: The stage transition uses a fixed and hand-picked 0.9 threshold. How sensitive are completion and precision to this value, and did you try progress-based or confidence-weighted switching that might be less brittle in cluttered or dynamic cases?
>
>
> We study the stage-transition threshold on InterPlay-Hybrid by sweeping it from 0.8 to 0.95. Performance varies smoothly between 0.8 and 0.9 and peaks at 0.9, while only the extreme setting 0.95 causes a clear drop in completion and precision.
>
> | Threshold |  Completion Rate(%) ↑ |  Sub-step Completion Ratio(%) ↑ |  Sub-step Precision(cm) ↓ |
> |----------|----------|----------|----------|
> | 0.80  | 47.9     |   65.6   |   13.5   |
> | 0.85  | 51.0     |   67.2   |  12.9   |
> | 0.90    | 53.8     | 71.8     | 11.2     |
> | 0.95   | 32.8     |   47.2   |   16.7   |
>
> Further, we did not introduce an extra progress- or confidence-weighted switching module, as this would require additional components to estimate such signals. Instead, as defined in Eqs. (11–16), the stage-switching signal is exactly the dense task reward, which is a continuous measure of how well the current sub-goal is being achieved. Its edge-wise RMD terms and adaptive aggregation weights already act as a confidence-weighted progress indicator, and we find this signal stable enough for a simple fixed threshold even in dynamic scenes.

---

### Official Review · Reviewer_WtPe · 2025-11-01

**Soundness:** 3
**Presentation:** 3
**Contribution:** 3
**Rating:** 6
**Confidence:** 3

**Summary:**

This paper introduces a novel framework for synthesizing physics-based human-object interactions (HOI) by leveraging Vision-Language Models (VLMs) to automatically generate goal states and reward functions. The core contribution is the Relative Movement Dynamics (RMD) representation—a fine-grained spatio-temporal bipartite graph that models part-level relationships between human and object components during interactions. This allows the VLM to reason about motion dynamics and generate semantically grounded, interaction-aware plans without manual reward engineering. The authors also contribute the InterPlay dataset, which includes long-horizon static and dynamic interaction tasks in diverse indoor scenes. Experiments in both single-task and multi-task settings demonstrate that the proposed method outperforms existing approaches in terms of completion rate, motion naturalness, and generalization.

**Strengths:**

1.	The introduction of Relative Movement Dynamics is a conceptually elegant and technically sound way to bridge high-level task instructions with low-level motion control. By modeling interactions as a bipartite graph of human and object parts with explicit motion trends, the method captures both spatial and temporal aspects of interaction in a unified manner.
2.	The method achieves state-of-the-art performance across multiple metrics (completion rate, sub-step precision) in both single-task and long-horizon multi-task scenarios. The improvements are especially notable in dynamic and hybrid interaction settings, where prior methods often fail.

**Weaknesses:**

1. The InterPlay dataset introduced in the paper includes articulated objects not found in current interaction datasets, but it does not specify what types of articulated objects are included or what kinds of interactions are involved.
2. The 6D pose of objects is crucial. How does the method prevent issues like unintended object rotation caused by inaccurate 6D pose estimation during interaction?
3. The construction of Graph B seems to be only briefly discussed in the paper, yet I believe the connectivity of this graph and the selection of edge weights would directly impact the generated motion outcomes.

**Questions:**

1. The paper claims the contribution of being the "first unified physics-based HOI synthesis framework leveraging the powerful world knowledge of VLMs," but it may not actually be the first work to use VLMs for physics-based HOI. This statement could be somewhat biased.

---

> ### Author Response · Authors · 2025-11-23
>
> We sincerely appreciate your thoughtful comments and constructive feedback.
>
> >W1: The InterPlay dataset introduced in the paper includes articulated objects not found in current interaction datasets, but it does not specify what types of articulated objects are included or what kinds of interactions are involved.
>
> As shown in **Lines 950-952** and demonstrated in our **video** (e.g., 3:35 - 3:50, 6:40 - 6:45): our dataset **includes doors** and **windows** modeled with **articulated joints**, and we define opening tasks on them. This goes beyond existing HOI-control datasets, where HITR only covers rearrangement tasks, and ScenePlan only considers interactions with static objects.
>
> ---
>
> >W2: The 6D pose of objects is crucial. How does the method prevent issues like unintended object rotation caused by inaccurate 6D pose estimation during interaction?
>
>
> Thank you for this question. In our current simulation-based framework, we leverage the ground-truth 6D pose provided by the simulator, which allows us to isolate the performance of our control policy from perception errors. For real-world deployment, we acknowledge that robust 6D pose estimation is critical. Our architecture's modularity allows for the direct integration of state-of-the-art 6D pose estimators. Furthermore, techniques like domain randomization and adversarial noise injection during training will be key to making our policy tolerant to pose estimation inaccuracies. We consider the integration of these robust perception and adaptation strategies a compelling direction for future work.
>
> ---
>
> >W3: The construction of Graph B seems to be only briefly discussed in the paper, yet I believe the connectivity of this graph and the selection of edge weights would directly impact the generated motion outcomes.
>
> The construction of Graph B is detailed in Lines 216-237 and Appendix A. B=(V,E,w) is a bipartite graph whose vertices V consist of human parts and object parts, and whose edges E and weights w are **instantiated directly** from the VLM planner’s output.
>
> We agree that the graph's connectivity and edge weights directly impact the motion outcomes. To systematically validate this, we conducted an ablation study on graph connectivity (Table 3), comparing our design against simplified variants ("multi-multi", "multi-one", "one-one"). The results confirm that our chosen connectivity is necessary for strong performance, as simplified graphs lead to a noticeable decrease. Regarding the edge weights, we selected the four semantic labels—separating, stationary, approaching, and unstable—as a deliberate trade-off between model complexity and performance. This set is compact enough for the VLM to predict reliably, yet sufficiently rich to encode the dominant relative motions (e.g., stationary for support, approaching for manipulation). The unstable label acts as a safety buffer for ambiguous cases. This design enables our policy to generate the diverse and physically plausible motions demonstrated in our experiments.
>
> ---
>
> >Q1: The paper claims the contribution of being the "first unified physics-based HOI synthesis framework leveraging the powerful world knowledge of VLMs," but it may not actually be the first work to use VLMs for physics-based HOI. This statement could be somewhat biased.
>
> We clarify that our novelty lies in the integrated use of a VLM for explicit task planning within a physics-based simulation for long-horizon, diverse HOI tasks. A careful comparison with prior work highlights key distinctions:
>
> **Anyskill [1]** uses CLIP (a vision-language model) to compute a reward signal, not for generative task or motion planning. It focuses on learning single manipulation skills in isolation, not on composing long-term interactions within a complex scene as our method does.
>
> **HumanVLA [2]** uses a standard EfficientNet for visual perception. It does not leverage the world knowledge or generative planning capabilities of a VLM to reason about and synthesize novel interactions from high-level instructions.
>
> **HOI-FHLI [3]** employs an **LLM-based** planner. Crucially, LLMs lack visual grounding; they cannot perform "motion imagination" by visually reasoning about object geometries, scenes, and potential contact points as our VLM can.
>
> Therefore, while related works use components of language or vision models, none integrate a visually-grounded VLM as a planner to drive a unified physics-based HOI framework across diverse, long-horizon tasks, which we demonstrate. We have revised the paper in response to your feedback, and the corresponding changes are highlighted in blue.
>
> ---
>
> **References**
> >[1] AnySkill: Learning Open-Vocabulary Physical Skill for Interactive Agents.
>
> >[2] HumanVLA: Towards Vision-Language Directed Object Rearrangement by Physical Humanoid.
>
> >[3] Human-Object Interaction from Human-Level Instructions.

---

### Author Response · Authors · 2025-12-03
**Final Remark**

Dear AC,

We sincerely appreciate the considerable effort you dedicated to reviewing our manuscript under these special circumstances. Unfortunately, we have not yet received further feedback due to the closed system. For your reference, we provide a brief summary of the reviewers' comments and outline how we have addressed their concerns.

We are encouraged that **all reviewers highlight the  technical soundness** of our framework. Reviewer call RMD “a **conceptually elegant and technically sound way** to bridge high-level instructions with low-level motion control” (WtPe), provides RL with a **direct mapping from plans to goal states and rewards** (xkXT), and **eliminates labor-intensive manual reward engineering** (7XuJ). They also appreciate the **generality** of our **unified long-horizon** HOI framework, supporting multi-stage interactions across diverse objects beyond many prior methods (xkXT, 7XuJ).

Reviewers endorse our experiments: InterPlay **fills a critical gap**, providing thousands of long-horizon, context-rich plans for **systematic multi-task HOI evaluation** (xkXT, 7XuJ). Reviewers also note **state-of-the-art** results in single- and long-horizon multi-task settings (7XuJ), with clear gains on dynamic/hybrid interactions (WtPe) and improved **motion naturalness/task alignment** backed by metrics and a user study (iGBJ).


Following the reviewers’ constructive suggestions, we addressed the key concerns in our rebuttal:

**(1) For planner reliability**, we added quantitative analyses of the RMD planner: measured error rates (xkXT, iGBJ), robustness under weaker planners/simplified prompts (xkXT, iGBJ), and part-decomposition evaluation on novel objects (7XuJ).

**(2) Further justification of model design**, we strengthened the justification of key choices via (i) a sweep of the stage-transition threshold (xkXT), (ii) connectivity ablations (Table 3) (WtPe), and (iii) a principled rationale for the 4-category edge weights (7XuJ).

**(3) For clarity**, we added a close-up supplementary video (xkXT), a runtime breakdown (offline planner vs. online controller) (xkXT), and clearer task scope/assumptions (iGBJ).

More broadly, we take this as a step toward **scalable, general-purpose HOI synthesis**: RMD is a **structured interface** that turns intent and scene context into **executable long-horizon interactions**, bridging VLM planning and RL control without task-specific demos or manual reward tuning. With InterPlay, we aim to move beyond bespoke pipelines toward **compositional, multi-task agents**.

We thank the reviewers and AC again for their time and constructive feedback.

Kind regards,

The authors

---

### Meta-Review · Area_Chair_uQ7n · 2025-12-27

**Summary:**

**Paper Summary**

The paper proposes a new human-object interaction (HOI) framework that enables long-horizon interactions with diverse objects. An associated dataset, Interplay, is collected and developed. Additionally, the Relative Movement Dynamics (RMD) is a fine-grained spatio-temporal representation designed for reward generation. The policies trained with RMD representation and guidance from other VLM planners are examined on the proposed framework. The experimental results suggest that the RMD representation can significantly boost policy performance on long-horizon HOI tasks.

---

After reading the paper, review comments, and author responses, the AC summarizes the paper's strengths and weaknesses below.

**Strengths**
- Significant Development Effort: This paper demonstrates substantial effort in developing a novel HOI platform. It allows researchers to test long-horizon tasks through both the simulated environments and the associated Interplay dataset.

- Methodological Contribution: The paper proposes a method featuring a new RMD representation and provides a thorough comparison against various existing solutions.

- Clarity and Presentation: The paper exhibits high clarity in general. It is self-contained and easy to follow, which allows readers to capture the core ideas quickly.


**Weaknesses**
- Limited Exploration of RMD Generation: There is insufficient study regarding how the RMD representation is generated. Specifically, the generation process relies heavily on a single VLM, GPT-4V, which may bound the results to that specific model's capabilities. Furthermore, the authors do not discuss alternative ways to generate the RMD representation.

- Scope of the Framework: The current framework is limited to single-agent tasks. It does not yet consider multi-agent collaborations or tasks involving complex object relationships.

- Incremental Technical Contribution: While the AC appreciates the extensive effort involved in developing the framework and the RMD representation, the technical contribution feels somewhat incremental.

- Efficiency Concerns: Some reviewers raised concerns regarding the inference time and the computational costs introduced by the proposed methods.

**Reviewer Concerns:**

The reviewers shared a common concern regarding the method's heavy reliance on GPT-4V, particularly in failure rates and numerical performance in object part decomposition. In response, the authors provided additional ablations to address these questions. They also acknowledged the current limitation to single-agent tasks and proposed potential ways to extend the framework. The AC feels that the rebuttal adequately addressed or discussed most of the reviewers' concerns.

However, the AC believes that the question of how to extend the framework and method to real-world applications or other simulated environments has not been discussed sufficiently. For instance, the VLM requires top-view images of the environments, which are often inaccessible in practice, particularly in multi-room scenes. Furthermore, generating rewards based on multiple goal states and object part knowledge is a highly unstable process in real-world applications.

Overall, the paper makes sufficient contributions by providing a unified HOI framework that contains challenging long-horizon tasks. However, the work would be more solid if the authors included a discussion on how to extend the components developed in the paper to real-world applications or more diverse situations. Given the positive recognition from the reviewers, the AC suggests accepting this paper under the following conditions:
- The camera-ready version should include a paragraph discussing how to extend the framework and the proposed method into real-world applications.
- The authors will provide a specific plan for the release of all assets, including the evaluation code and the Interplay dataset, as these resources were guaranteed during the rebuttal process.

**Reviewer Scores:**

The paper's initial scores were [6, 6, 8, 8], indicating a unanimously positive evaluation. The authors also offered detailed responses to the reviewers' concerns. While a full discussion might have seen one or two reviewers raise their scores from 6 to 8, the AC feels the submission still sits between borderline and weak accept (poster) due to the issues noted above.

---

### Decision · Program_Chairs · 2026-01-26

Accept (Poster)